

# HydroSat: a repository of global water cycle products from spaceborne geodetic sensors

Mohammad J. Tourian[1], Omid Elmi[1], Yasin Shafaghi[2], Sajedeh Behnia[1], Peyman Saemian[1], Ron Schlesinger[1], and Nico Sneeuw[1]

[1]University of Stuttgart, Institute of Geodesy (GIS), Stuttgart, Germany
[2]GAF AG, Munich, Germany

**Correspondence:** Mohammad J. Tourian (tourian@gis.uni-stuttgart.de)

**Abstract.** Against the backdrop of global change, both in terms of climate and demography, there is a pressing need for monitoring the global water cycle. The publicly available global database is very limited in its spatial and temporal coverage worldwide. Moreover, the acquisition of in situ data and their delivery to the database are in decline since the late 1970s, be it for economical or political reasons. Given the insufficient monitoring from in situ gauge networks, and with no outlook for

improvement, spaceborne approaches have been under investigation for some years now. Satellite-based Earth observation with its global coverage and homogeneous accuracy has been demonstrated to be a potential alternative to in situ measurements. This paper presents HydroSat as a repository of global water cycle products from spaceborne geodetic sensors. HydroSat provides time series and their uncertainty of: water level from satellite altimetry, surface water extent from satellite imagery, terrestrial water storage anomaly from satellite gravimetry, lake and reservoir water storage anomaly from a combination of

satellite altimetry and imagery, and river discharge from either satellite altimetry or imagery. These products can contribute to understanding the global water cycle within the Earth system in several ways. They can act as inputs to hydrological models, they can play a complementary role to current and future spaceborne observations, and they can define indicators of the past and future state of the global freshwater system. The repository is publicly available through http://hydrosat.gis.uni-stuttgart.de.

## 1 Introduction

To understand the global hydrological cycle and the Earth system in general, measurements are needed to estimate storages and fluxes on a spatial scale from local to continental, and on a temporal scale sufficient to resolve even diurnal variations (Lettenmaier, 2006). However, current knowledge of spatial and temporal dynamics of water storage and fluxes on landmasses is limited (Alsdorf and Lettenmaier, 2003). The water surface elevation variation and bathymetry of rivers and lakes are not

sufficiently known. The depth of soil moisture is not known on a global scale. Rain gauge measurements are spatially not dense enough to represent the input to the hydrological cycle. The number of discharge gauge stations that contribute to the global database has been declining steadily over the past decades. Water storage estimates from traditional in situ measurements are





subject to large uncertainty (Alsdorf and Lettenmaier, 2003; Strassberg et al., 2007; Yeh et al., 2006; Rodell et al., 2006; Rieg-ger et al., 2012). In fact, by today's knowledge, storage, fluxes, and their changes over time cannot be quantified properly. So

they still remain our known unknowns (Famiglietti, 2012).

Spaceborne geodetic sensors, designed for a variety of purposes, have established themselves as valuable tools for oceano-graphic, cryospheric and also hydrological applications (Alsdorf et al., 2007). Satellite altimetry, originally aiming at oceanog-raphy and geodesy, has demonstrated its potential as virtual lake and river gauges (Alsdorf et al., 2007; Papa et al., 2010; Berry et al., 2005). The exciting possibility of surface water extent monitoring using satellite imagery (optical and SAR) raises hopes

of better capturing surface water variability and providing a realistic overview of hydrologic behavior at the basin scale. Since 2002 the satellite mission Gravity Recovery And Climate Experiment (GRACE) has been providing a fundamentally new re-mote sensing tool for a wide spectrum of Earth science applications (Tapley et al., 2004). GRACE is able to monitor changes in the ocean and the global hydrological cycle through measuring changes in the Earth's gravitational field from space.

The aforementioned missions promote novel approaches in oceanography, geophysics, hydrology and hydro-meteorology.

Among these sciences, there is an urgent need for more observational data, particularly in hydrology. This necessity arises from the above mentioned limited knowledge of the spatial and temporal dynamics of the freshwater variations and fluxes (Sneeuw et al., 2014). Given such a pressing need, spaceborne geodetic sensors, with their global coverage and homogeneous accuracy, are viable choices over *in situ* measurements. It should be noted, however, that despite being the only source of information in data-poor regions, satellites do not yet provide the desired spatio-temporal resolution. The spatio-temporal resolution of

spaceborne sensors is defined by their orbital characteristics and their measurement concept. Satellite altimetry missions are typically placed in repeat orbits with a given number of revolutions within a certain number of days e.g. 35 days for ENVISAT, 10 days for Jason series and 27 days for Sentinel-series (Fu and Cazenave, 2001). GRACE can provide meaningful signals at its best spatial resolution on the monthly time scale. The spatio-temporal resolution problem is not as pronounced for optical and SAR satellite imaging missions because they provide images with a relatively high spatial resolution (about 250 m for

MODIS and 30 m for Landsat) at an acceptable temporal resolution. However, cloud cover is a limiting factor in using the optical images for generating a dense time series of surface water area.

Space-based water cycle monitoring is entering a new era in view of the wealth of present and future missions. Satellite altimetry has already been put on an operational basis by the Sentinel-3 satellites of the European Copernicus program. At the same time, research satellites such as CryoSat-2, SARAL/AltiKa and Jason-3 remain in orbit and provide complementary

space-time measurements. On the other hand, MODIS sensors on NASA's Terra and Aqua satellites have been acquiring medium resolution satellite images daily since 2000. In addition to the MODIS images, high resolution optical and SAR satellite images are available from Landsat 8 and Sentinel 1, 2. Moreover, the planned SWOT mission, due for launch in 2022, will represent a paradigm change in monitoring surface water, providing a 2-D swath as opposed to conventional 1-D profiling. SWOT aims to monitor surface water elevation, extent, slope and also river discharge for all rivers wider than

100 m (Biancamaria et al., 2016). In addition, GARCE Follow-On was launched in 2018 to ensure continuity of the GRACE mission after its successful 15-year monitoring of water storage variation. The current constellation addresses many existing limitations and opens a significant area of investigations into the operational use of satellites for hydrological purposes.



Inspired by the increasing need to monitor the global water cycle and by the potential offered by the existing constellation, the HydroSat repository was initiated in 2016. HydroSat hosts global water cycle products from spaceborne sensors: 1) surface water extent of lakes and rivers, 2) water level of inland water bodies, 3) water storage anomaly of hydrological basins, lakes and reservoirs, and 4) river discharge for large and small rivers. These products are the results of research studies and projects on the application of spaceborne geodetic sensors for hydrology conducted at the Institute of Geodesy (GIS), University of Stuttgart. This paper describes HydroSat data products with a detailed explanation of the algorithm behind them: water level time series from satellite altimetry in Section 2, river width estimation from satellite imagery in Section 3, water storage anomaly for hydrological river basins, lakes and reservoirs in Section 4 and, finally, river discharge estimation from satellite altimetry and imagery in Section 5. For each data product, some representative examples are given to support the product description.

## 2   Inland water level from satellite altimetry

In the last three decades, satellite altimetry has been used as a monitoring tool for inland water surfaces and the hydrological cycle. In particular, monitoring water level of large rivers and lakes has been the goal of research since the launch of the TOPEX/Poseidon and Envisat missions (Table 1) (Birkett, 1995; Crétaux et al., 2011). The use of satellite altimetry for inland water monitoring has been facilitated by the advent of two different developments:

– Open-Loop Tracking Command (OLTC) (used in missions with gray background in Table 1)

– Operation in Synthetic Aperture Radar (SAR) mode (implemented in missions with bolded text in Table 1)

The OLTC was first implemented on the Poseidon-3 altimeter onboard the Jason 2 satellite to improve the on-board tracker through properly setting the reception window of the return echoes. This can be realized by OLTC tables consisting of the overflown surface elevation (Le Gac et al., 2019) Such an adjustment helps the altimeter to successfully track inland water bodies, especially within rough topography. After the successful experience with Jason-2, OLTC was also implemented in SARAL/AltiKa, Jason-3 and Sentinel-3. On the other hand, inland water monitoring using satellite altimetry has benefited extremely from the Delay Doppler technique or the concept of Synthetic ApertureRadar (SAR), first used in CryoSat-2. In the delay Doppler technique, after delay compensation, the height estimates are sorted by Doppler frequencies and integrated in parallel, thus accumulating more looks than a conventional altimeter and a relatively small along-track footprint (ca. 250 m) (Raney, 1998). Sentinel-3A was the first mission to operate in SAR mode and in OLTC nearly globally.

Inspired by recent developments, the abundance of altimetry missions, and the SWOT mission in view, the development of repositories and services to provide inland water level time series to supply data for Earth system understanding, hydrologic cycle monitoring, and hydraulic studies is becoming more important than ever. Table 2 lists currently available websites or repositories for providing water level time series of inland water bodies.

Hydroweb was the first website providing altimetric water level time series. The website, now hosted in THEIA, is a CNES project, which was initiated and monitored by LEGOS in 2003 based on all existing and past altimetry missions,

**Table 1.** Satellite altimetry missions from 1985– and their characteristics

| mission | operated by | life time | height [km] | inclination [°] | rev./day[1] | frequency [GHz] |
|---------|-------------|-----------|-------------|-----------------|-------------|-----------------|
| Geosat | NOAA | 03.1985–09.1989 | 785 | 108.0 | 244/17 | 13.5 |
| ERS-1 | ESA | 07.1991–03.1996 | 785 | 98.5 | 501/35 | 13.5 |
| TOPEX/Poseidon | CNES, NASA | 09.1992–01.2006 | 1336 | 66.0 | 127/10 | 13.6 & 5.3 |
| ERS-2 | ESA | 04.1995–09.2011 | 781 | 98.5 | 501/35 | 13.5 |
| GFO | US-Navy | 02.1998–11.2008 | 784 | 108.0 | 244/17 | 13.5 |
| Jason-1 | CNES, NASA | 01.2002–06.2013 | 1336 | 66.0 | 127/10 | 13.6 & 5.3 |
| ENVISAT | ESA | 03.2002–04.2012 | 800 | 98.5 | 501/35 | 13.5 & 3.2 |
| Jason-2 | CNES, NASA, NOAA, EUMETSAT | 06.2008–ongoing | 1336 | 66.0 | 127/10 | 13.6 & 5.3 |
| HY-2A | NSOAS | 08.2011–ongoing | 971 | 99.3 | 193/14 | 13.58 & 5.25 |
| ICeSat | NASA | 01.2003–10.2009 | 600 | 94.0 | 2723/183 | Laser: 1064 & 532 nm |
| **CryoSat-2** | **ESA** | **04.2010–ongoing** | **717** | **92.0** | **5344/369** | **13.575** |
| SARAL/AltiKa | ISRO, CNES | 02.2013–ongoing | 800 | 98.5 | 501/35 | 35.75 |
| Jason-3 | CNES, NASA, NOAA, EUMETSAT | 01.2016–ongoing | 1336 | 66.0 | 127/10 | 13.6 & 5.3 |
| **Sentinel-3A** | **ESA, GMES** | **02.2016–ongoing** | **815** | **98.6** | **385/27** | **13.6 & 5.3** |
| **Sentinel-3B** | **ESA, GMES** | **04.2018–ongoing** | **815** | **98.6** | **385/27** | **13.6 & 5.3** |
| ICeSat 2 | NASA | 09.2018–ongoing | 480 | 92.0 | 1387/91 | Laser: 1064 & 532 nm |
| HY-2B | NSOAS | 10.2018–ongoing | 971 | 99.3 | 193/14 | 13.58 & 5.25 |
| **Sentinel-6 Michael Freilich** | **EUMETSAT, NASA** | **11.2020– ongoing** | **1336** | **66.0** | **127/10** | **13.5** |
| SWOT | NASA,CNES | 02.2022–02.2025 | 890 | 77.6 | 292/21 | 35.75 |

[1] Nodal day

from TOPEX/Poseidon until Sentinel-3B. A substantial portion of water level time series of lakes and rivers are provided in Near Real Time (NRT). CNES, LEGOS, and CLS have further extended their monitoring services under agreements with Copernicus. As a results, they provide water level time series of 94 selected lakes within the context of the Copernicus Climate Change Service, the so-called C3S LWL. They further provide historical and NRT water level time series of several lakes and rivers through the VITO Earth Observation portal. In 2009, in a cooperation of ESA and De Monfort University, the River&Lake website became available. On this website, which is no longer maintained, Global NRT products were available. Later in 2013, the Database for HydrologIcal Time Series of Inland waters (DAHITI) was developed by the Deutsches Geodätisches Forschungsinstitut at the Technical University Munich (DGFI-TUM) to provide water level time series of inland waters (Schwatke et al., 2015a). DAHITI provides a variety of hydrologial information on lakes, reservoirs, rivers, and wetlands derived from different satellite missions. Since 2017, the Global Reservoirs and Lakes Monitor (G-REALM) provides time series of water level variations for some of the world's largest lakes and reservoirs. Unlike G-REALM, the Global River Radar Altimeter Time Series (GRRATS) focuses on rivers and provides water level time series over 39 rivers spanning the time period 2002–2016 using Envisat-series and Jason-series (Coss et al., 2020). Within a rather unprecedented framework, the



**Table 2.** Providers of water level time series from satellite altimetry

| Product | operated by | source | Remark |
| --- | --- | --- | --- |
| Hydroweb | CNES | http://hydroweb.theia-land.fr | NRT available for some lakes and rivers |
| River& Lake | ESA | http://altimetry.esa.int/riverlake | no longer maintained |
| DAHITI | Deutsches Geodätisches Forschungsinstitut (DGFI), TU Munich | https://dahiti.dgfi.tum.de Schwatke et al. (2015a) | Kalman filter approach |
| HydroSat | Insititute of Geodesy University of Stuttgart | http://hydrosat.gis.uni-stuttgart.de | Hi-Rate products are available |
| G-REALM | United States Department of Agriculture | https://ipad.fas.usda.gov/cropexplorer/global_reservoir | Lakes and reservoirs only |
| GRRATS | The Ohio State University | https://doi.org/10.5067/PSGRA-SA2V1 Coss et al. (2020) | Envisat- and Jason-series over 39 rivers |
| AltEx | USAID and NASA | https://altex.servirglobal.net/ Markert et al. (2019) | web application for exploring altimetry data Jason-2, Jason-3 and Saral/AltiKa |
| C3S LWL | CLS on behalf of Copernicus and European Commission | https://doi.org/10.24381/cds.5714c668 | 94 selected lakes are available |
| Water Level on VITO | Copernicus Global Land Operations CNES, LEGOS, and CLS | https://land.copernicus.eu/global/products/wl | NRT time series are available |

open source web application, AlteEx allows for exploring altimetry datasets and generating water level time series on-the-fly. AltEx is supported by the U.S. Agency for International Development (USAID) and NASA (Markert et al., 2019; Okeowo et al., 2017).

In HydroSat, water level time series are provided in two modes: Standard-Rate (SR) and High-Rate (HR). The HR products come with an improved temporal resolution relying on multi-mission altimetry for both lakes and rivers. In the following a detailed description of SR and HR products will be provided.

## 2.1 Standard-Rate water level time series from satellite altimetry

Whenever the satellite ground track crosses a hydrological object, a so-called Virtual Station (VS) can be determined. Boundaries of a VS are typically defined based on the type of the water body and the disposition of the altimetry track over the object. All measurements inside a lake or reservoir, for instance, could belong to the same virtual station. This is to assume that the along-track geoid height and altimetric corrections are properly known. In case of a river, the assumption further implies that





the spatial dynamics of the water body is locally negligible. Defining a VS, in effect, allows for reducing random noise carried
by the altimetric measurements.

HydroSat follows a rather flexible approach in defining a VS. The boundaries can be determined using static or dynamic
shapefiles, a radial extent around a specific point of interest, or an intersection of the two. Furthermore, HydroSat employs
auxiliary sources of information like the water occurrence frequency derived from Landsat imagery by Pekel et al. (2016). This
allows for removing measurements over areas with a water occurrence frequency below a certain threshold. Figure 1 shows
the block diagram for generating SR water level time series, where the properties of the VS is the basic setup information to
be introduced. Defining the VS setup is in fact the only subjective choice made throughout the whole procedure. Nevertheless,
flexibility in defining a VS is tolerated as long as reproducibility of the results is guaranteed.

The altimetric water level is initially determined for each sample within the VS. First, range measurements are corrected for
geophysical effects (solid earth tide and pole tide) and path delays caused by the atmosphere (wet tropospheric, dry tropospheric
and ionospheric). The water level is then calculated by subtracting the corrected range from the satellite altitude. In the next
step, the reference height is changed to geoid according to static gravity field models from XGM2019e (Pail et al., 2018),
EGM2008 (Pavlis et al., 2012), or EIGEN6C3 (Förste et al., 2012). To ensure a robust estimation at each overpass, the median
of orthometric heights inside the VS is chosen to be the representative height.

It is important to notice that regardless of the choice of the VS, retracker, corrections, and statistics by which a typical
altimetric water level time series is generated, the result may be affected by many outliers. As already mentioned, missions
supporting OLTC and SAR mode are less likely to provide erroneous range measurements. However, neither the OLTC nor the
delay/Doppler concept are capable of compensating for all unwanted radar processes over inland water bodies. Their usefulness
may even be restricted by a number of known factors – e.g. crossing angle for SAR missions. An outlier identification algorithm
is, therefore, required to clean the final water level time series from erroneous measurements.





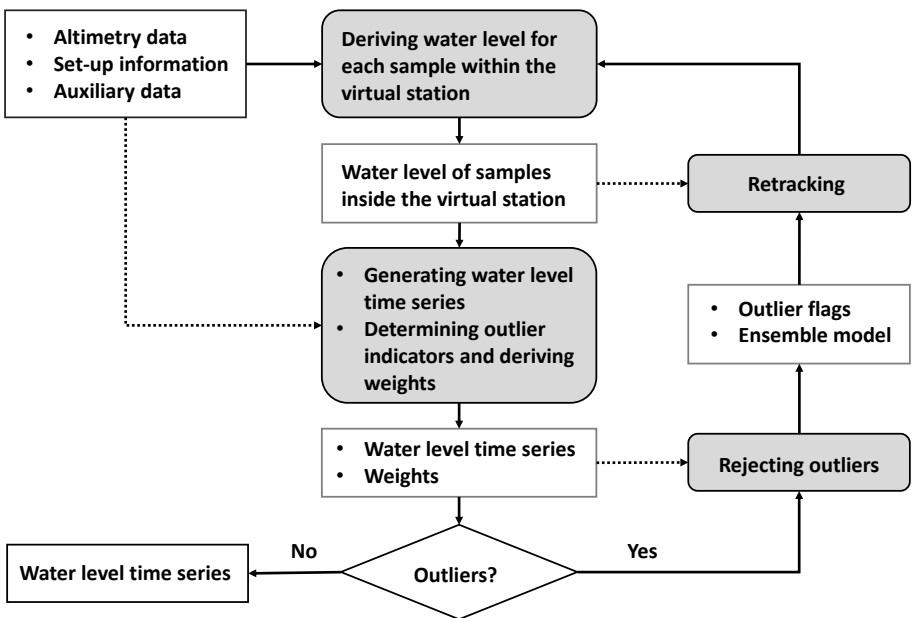

**Figure 1.** Flowchart of obtaining SR altimetric water level time series for a single virtual station.

In order to identify the outliers, HydroSat uses an automated, data-driven outlier identification methodology designed within an iterative, non-parametric adjustment scheme. As indicated in Figure 1, the inputs to the algorithm are the water level time series and stochastic information derived from a number of outlier indicators (e.g. standard deviation of along-track estimates within a VS). The algorithm uses Singular Spectrum Analysis (SSA) for gap-filling, Savitzky-Golay filtering for smoothing, and a specially developed outlier identification method (Behnia et al., 2021b). The identification method benefits from applying a local kernel derived based on a local definition of an outlier. The identified outliers are not discarded instantly. The overall scheme allows for a *possible* correction of the outlying estimation through a retracking method based on Leading Edge Identification with Prior Information (LEIPI) (Behnia et al., 2021b). Such possibility is assessed via comparing the single water level estimations inside the VS with the water level estimation coming from an ensemble model. The model, which is a by-product of the outlier identification algorithm, is used to bound the search area for identifying the true leading edge of the waveform. After retracking, the newly estimated heights are verified by their similarity to the along-track pattern of outlier-free cycles.

Figure 2 shows a representative collection of SR water level time series over Lake Bankim in Cameroon and rivers Mississippi in US, Sao Francisco in Brazil, Karun in Iran, Yangtze in China, and Irrawaddy in Myanmar. If available, the time series are compared against in situ measurements and altimetric water level time series from other sources. One shall notice that AltEx, C3S LWL, and VITO Water Level are excluded from this comparison. In case of AltEx, any comparison would have been subjective as the quality of the water level time series are dependent on the geometric choice of a virtual station by the user. As for the C3S LWL and VITO Water Level, we have observed no difference between these time series with





those from Hydroweb (the same argument holds for objects shown in 2.2.1). For the rivers Sao Francisco (bottom left) and Mississipi (middle left), the altimetric heights agree with the in situ measurements with correlation coefficients of 0.99 and

0.97, respectively. In case of the Karun river (top right), the correlation falls off to 0.85 which is satisfactory given the 250 m crossing width, mountainous topography, and high seasonality of its semiarid climate. The SR water level time series are well representative of water level dynamics, an example being the case of the River Irrawaddy (bottom left) showing the high rank correlation of 0.91 with in situ discharge measurements. The results over the Yangtze River (middle right) and Bankim Lake (top left) agree well with DAHITI, Hydroweb and G-REALM. Over these water bodies — like many other water bodies around

the world — no in situ measurements are publicly available, highlighting the critical contribution of spaceborne measurements and the existing repositories.

It is important to notice that after rejecting or correcting the outlying measurements, HydroSat does not low-pass filter the water level time series. It can therefore fully capture the high frequency behavior within the limitations of satellite sampling. However, this strategy may come at the cost of a higher error level for some stations, e.g. the Mississippi river.

**Figure 2.** Standard-Rate altimetric water level time series over Lake Bankim in Cameroon and rivers Mississippi in US, Sao Francisco in Brazil, Karun in Iran, Yangtze in China, and Irrawaddy in Myanmar.



A SR water level time series is the basic altimetry product of HydroSat. It is the input to algorithms which provide HR water level time series over lakes, reservoirs, and rivers.

## 2.2   High-Rate water level time series from satellite altimetry

In order to obtain a HR product and cope with the limitation of temporal sampling of single-satellite inland water monitoring, multi-mission altimetry is applied. The multi-mission altimetry for lake monitoring is now a standard approach practiced by
various studies and data providers (Crétaux et al., 2013). Assuming that a lake surface is an equipotential surface, allows to perform even calibration studies over lakes. However, for multi-mission altimetry one challenge is posed by the inter-satellite biases, which impedes a straightforward combination of water level measurements. Moreover, inaccurate atmospheric corrections (wet tropospheric) may cause large biases of several decimeters, which is even more pronounced for rivers due to inhomogeneous neighboring topography (Fernandes et al., 2014). Unlike lakes, multi-mission studies over rivers are very
limited. Only a few studies have been dedicated to water level monitoring of rivers using a multi-mission approach with the focus on improving the temporal resolution (Tourian et al., 2016; Boergens et al., 2017). Here the challenge is to combine measurements from different missions at different locations with dissimilar dynamic behavior and hydraulic parameters. In general, HydroSat follows two different approaches for obtaining the HR product over lakes and rivers, outlined in the following sub-sections.

### 2.2.1   High-Rate altimetric water level over lakes and reservoirs

HR lake and reservoir level time series are provided based on the well-known multi-mission concept. As already mentioned, however, the integration of single water level time series is hampered by unknown biases, typically referred to as inter-satellite biases. Studies have been performed to tackle the problem, some targeting the specific case of lakes and reservoirs. Crétaux et al. (2009) provide an overview of research studies aiming at quantifying the absolute altimeter bias of each satellite. They
also establish an absolute calibration site over the Lake Issyk-Kul in central Asia for estimating the absolute bias of altimetry satellites and retrackers. Bosch et al. (2014) conduct a global cross-calibration analysis over the oceans to estimate the so-called radial bias – defined as the overall error of an altimeter system – in a relative manner. These results are interpolated by Schwatke et al. (2015b) and applied over inland altimetry water level time series. In order to minimize other sources of bias, DAHITI applies identical retracker and geophysical corrections to all measurements. Wang et al. (2019) correct for inter-
satellite biases based on a maximum likelihood approach either for two missions with overlapping periods or by introducing ICESat-2 as intermediary for non-overlapping ones.

HydroSat resolves biases in a relative, generic, regional and mission independent manner (see Figure 3). First, SR water level time series are categorized into groups of temporally overlapping and non-overlapping time series. For a group of overlapping time series, relative biases are estimated by minimizing a cost function for the merged time series. The cost function represents
the difference between the power content of individual SR time series. If stationarity holds, minimizing such a cost function ensures the estimation of a correct relative bias. If stationarity does not hold, or in case any unresolved bias remains (scenario 2), remotely sensed surface area time series are used to act as an anchor of biased time series, allowing for estimation of


the relevant biases. Here, a 2D cost function in surface area-water level coordinates is minimized within a Gauss-Helmert adjustment scheme. Such a cost function is also used in case measurements have no overlap (scenario 1), leading to bias

removed SR lake water level time series. It is important to notice that relative biases are not necessarily estimated between missions but between time series. This allows for considering the inaccuracies of geoid or altimetry corrections within one lake and over different tracks (Behnia et al., 2021a).

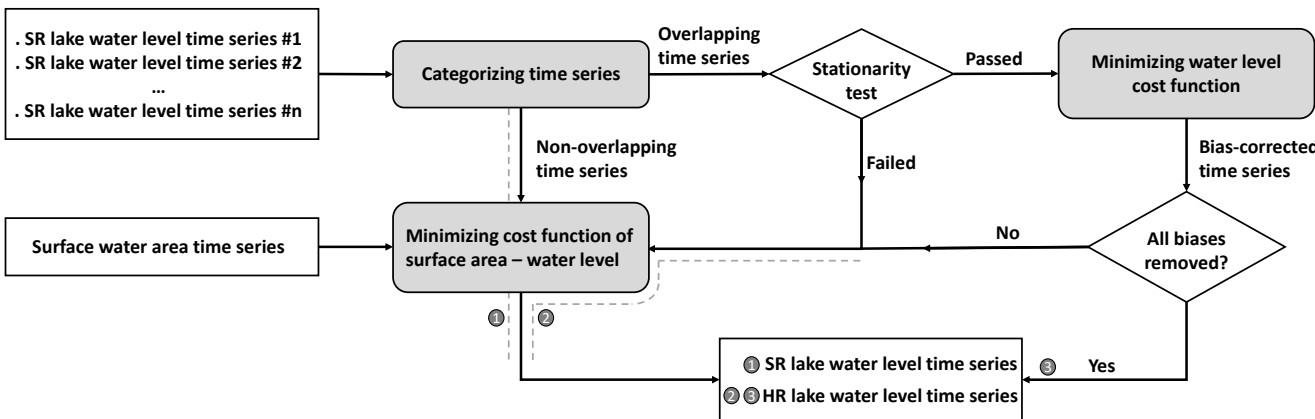

**Figure 3.** Flowchart of bias correction for obtaining SR and HR altimetric water level time series over lakes.

A relative solution is preferred because absolute estimation of biases requires along-track in situ measurements. The absolute inter-satellite biases over a specific region, on the other hand, are not necessarily applicable elsewhere due to inhomogeneity

of correction models at global scale. It shall also be mentioned that a great portion of lakes and reservoirs are monitored by a few altimetry missions, often times with less than sufficient periods of overlap. For instance, the long-term, overlapping, 10-day-revisiting Jason series, only monitor a small number of lakes given their coarse ground-track pattern. Our proposed method is therefore designed to be least affected by these restricting conditions.

Figure 4 shows HR water level time series over a selection of five lakes and reservoirs of very diverse characteristics.

While lake water level time series from HydroSat, DAHITI, Hydroweb, and G-REALM are in acceptable agreement with one another and with in situ measurements, some differences are noticeable. Over lake Erie, for instance, HydroSat better captures measurements at the tails of the water level distribution meaning that the actual fluctuations in lake level are better presented. This is due to the fact that besides outlier rejection, as described in Section 2.1, no further smoothing is applied. The HR water level time series of lake Urmia signals yet another difference. All altimetric time series seem to have captured the long-term

depletion of the lake level, followed by the recent and ongoing restoration period (Saemian et al., 2020). The majority, however, overestimate the lake level between late 2010 and early 2019. The altimetric measurements during this period are mainly from Jason-2 and Jason-3 missions. The satellite's ground track happens to cross the shallow south east of the lake. During this 9-year low water period, the altimeters have measured the range to the salt pan that remain in the southern part after the lake





desiccation. Excluding such a measurement without incorporating auxiliary sources of data is rather impossible. HydroSat can

deal with this issue within the inter-satellite bias estimation (Figure 3) by taking the surface area into account. Any SR water

level time series that exhibits inconsistency with an expected behavior fails to contribute to the HR lake water level time series.



**Figure 4.** High-Rate altimetric water level time series over lakes Ngoring in China ($610 \, \mathrm{km}^2$), Tana in Ethiopia ($2156 \, \mathrm{km}^2$), Urmia in Iran ($5200 \, \mathrm{km}^2$), Sobradinho in Brazil ($4214 \, \mathrm{km}^2$) and Erie in North America ($25744 \, \mathrm{km}^2$)


### 2.2.2  High-Rate altimetric water level over rivers

HR water level time series are obtained over rivers, by a method developed by Tourian et al. (2016), in which SR time series from individual altimetry missions are merged. This method allows combining all VSs from multiple satellite altimeters along

a river hydraulically and statistically. As an example, Figure 5 shows individual SR time series from different satellite missions along the Weser River in Germany. The idea of a HR product is to combine all these measurements for an arbitrary location along the river into a time series with improved temporal resolution.

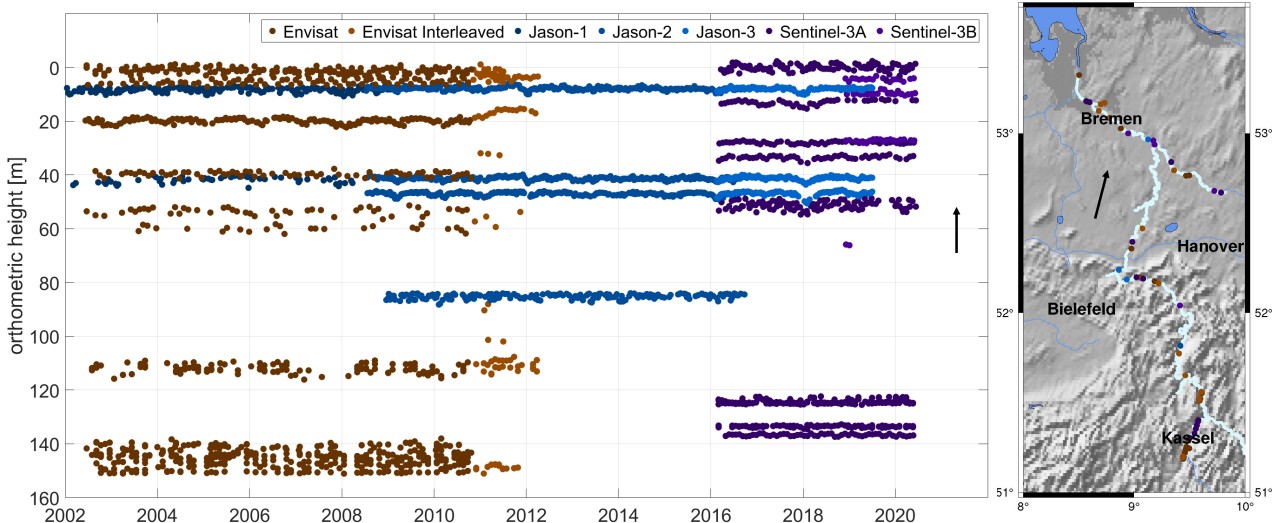

**Figure 5.** Water level estimates from different satellite altimetry missions along the Weser River in Germany.

Figure 6 shows the main processing steps of the densification method. Initially, the time lag due to streamflow between the altimetric virtual stations and the selected location is estimated. The average river width from satellite imagery together

with the slope derived from satellite altimetry are used as inputs to a simple empirical hydraulic equation, which ultimately estimates the average flow velocity and thus the time lag (Tourian et al., 2016; Bjerklie et al., 2003)

Using the estimated time lag, the water level hydrographs of all measurements are shifted and stacked at the selected location. The stacked time series is then normalized according to its statistical distribution with the 3rd percentile assigned to 0 and the 85th percentile to 1. Outliers are then identified and removed from the normalized time series by a Student's t-test for a sliding

time window of one month. All measurements outside the confidence limit are identified as outliers and removed from the measurements. The outlier-free normalized time series is then rescaled according to the statistical water level distribution of the selected site. Details on the implementation of the method can be found in (Tourian et al., 2016).

Figure 7 shows the HR altimetric water level over 6 selected rivers of small, intermediate and large size: Seine in France, Missouri in the USA, Congo in the Republic of Congo, Weser in Germany, Vistula in Poland, and Po in Italy. For the rivers with

no in situ water level time series, the HR water level time series are compared with in situ discharge data and rank correlations





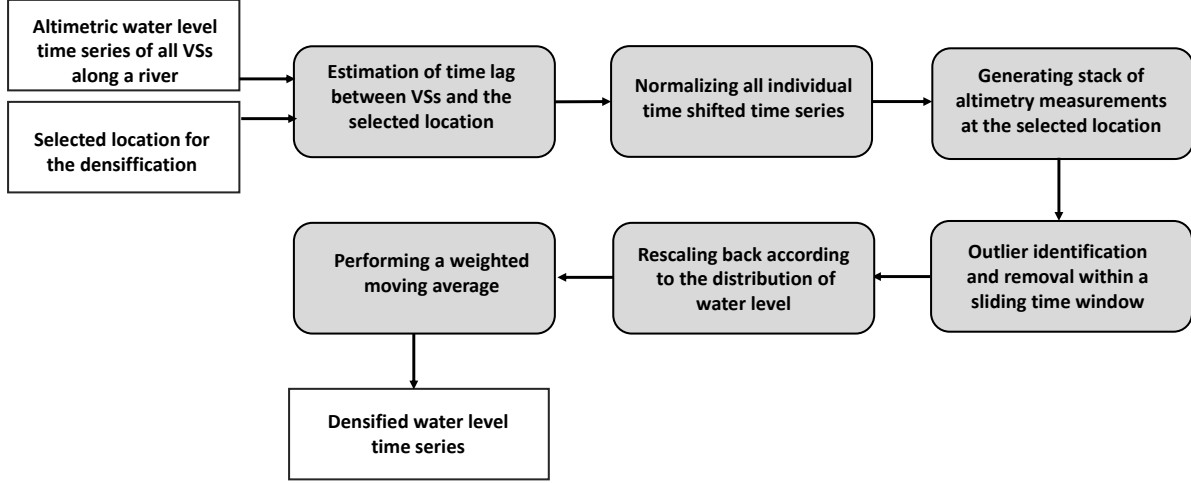

**Figure 6.** Flowchart of obtaining HR product over rivers through densification of individual altimetric water level time series along a river

are reported. It is worth mentioning that the Weser is a river with an average width of ca. 50 m and a maximum of ca. 150 m, over which the water level agrees with the discharge with a rank correlation coefficient of 0.71. Similarly, the Vistula River with its average width of 50 m is an utterly challenging river for satellite altimetry. However, the HR water level time series shows a rank correlation coefficient of 0.65 with in situ river discharge. Note that the large uncertainty in the HR time series is

due to the large discrepancy between measurements from different satellite missions along the river (see Figure 5).



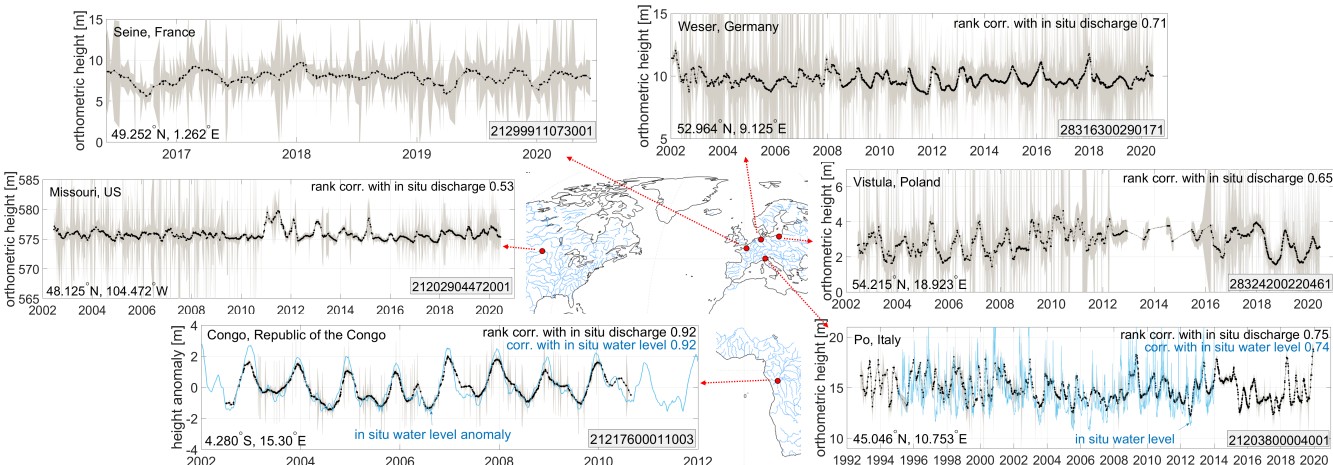

**Figure 7.** High-Rate altimetric water level over 6 selected rivers with different average width: Seine River (with average width of 170 m and 67 VSs) in France, Missouri River in US (with average width of 220 m and 61 VSs), Congo River in Republic of the Congo (with average width of 2700 m and 59 VSs), Weser River in Germany (with average width of 100 m and 47 VSs), Vistula River in Poland (with average width of 50 m and 31 VSs) and Po River in Italy (with 350 m average with and 46 VSs). HydroSat ids are provided in the figures bottom left.

## 3 Surface water extent from satellite imagery

Surface water storage is an important component of the hydrological cycle and its accurate monitoring requires a realistic representation of the surface water extent (Elmi et al., 2016). Lack of such observations, however, has obscured the proper quantification of the freshwater storage and its spatio-temporal dynamics over many water bodies. With their global coverage and fine temporal resolution, satellite images provide an opportunity to monitor the surface water extent on a global scale and for almost all river basins. To this end, attempts have been made to generate dynamic water masks from different spaceborne missions with various temporal and spatial resolution. Some studies (Klein et al., 2017; Zhang and Gao, 2016; Khandelwal et al., 2017) take advantage of MODIS images to generate time series of surface water mask with a fine temporal resolution. Due to the coarse spatial resolution of MODIS images, however, small water bodies are excluded from their dataset. On the other hand, some studies (Donchyts et al., 2016; Pekel et al., 2016; Schwatke et al., 2020) use Landsat images to generate time series of surface water bodies. In comparison to MODIS, Landsat images have a better spatial resolution (30 m). Nevertheless, their coarse temporal resolution is a limiting factor for monitoring the fast dynamics of the water bodies.

Similar to altimetric water level time series, Hydroweb is the first website to provide lake surface area time series (Crétaux et al., 2011). Recently DAHITI (Schwatke et al., 2019) has also boosted its database by the time series of lake area from optical satellite images (Table 3). Moreover, since recent years the Bluedot observatory provides reliable and timely information about surface area of lakes and reservoirs based on Sentinel-2 imagery, globally.



**Table 3.** List of sources for providing time series of surface water extent from satellite imagery

| Product | operated by | source | Remark |
|---|---|---|---|
| Hydroweb | CNES | http://hydroweb.theia-land.fr | available for lakes |
| DAHITI | Deutsches Geodätisches Forschungsinstitut (DGFI) | https://dahiti.dgfi.tum.de | available for lakes |
| HydroSat | Insititute of Geodesy University of Stuttgart | http://hydrosat.gis.uni-stuttgart.de | available over rivers and lakes |
| Bluedot observatory | Copernicus, European commission ESA, USGS, Amazon Web Services | https://blue-dot-observatory.com | available for lakes and reservoirs |

While over lakes and reservoirs dynamic water masks exist (Table 3), to the best of our knowledge, no specific dataset for dynamic river masks has been developed so far. Given the complexities in extracting river water masks from stacks of satellite imagery, most efforts have been limited to the development of static water masks. Allen and Pavelsky (2018) and Yamazaki et al. (2019), for instance, have extracted river water masks through stacks of satellite images. HydroSat provides time series of river reach area together with the dynamic lake and reservoir masks using optical satellite imagery.

A classic approach to extract water mask from optical images is to use pixel-based image segmentation algorithms, which are typically based on defining a threshold in the image pixel value histogram. While pixel-based algorithms are easy to implement, they fail to provide accurate water masks, especially over river reaches. The method performs well when the pixel values of the object and background create a combination of two normal distributions in the image histogram. This assumption, however, does not hold for the complicated river medium where the water body occupies only a small portion of the whole image. In fact, for a river reach, pixel values depend on several factors like water quality, roughness of water surface, chemical properties, load of sediments, vegetation canopy, and the depth of the water column (Elmi, 2019). Therefore, along the shorelines the complicated combination of water, vegetation, wet and dry soil within a pixel makes it almost impossible to find a unique threshold for distinguishing water from land. Region-based segmentation techniques, on the other hand, consider each pixel as an element of a larger region and use spatial information as well as pixel value to assign labels to the areas. Like other natural phenomena, water bodies show a strong spatial correlation in satellite images. Therefore, including contextual information can significantly improve the results water mask. Moreover, every pixel has a certain behavior during the time of monitoring, dominated by the seasonal cycle. Hence, in addition to spatial correlations, strong temporal correlations can be used as an additional source of information.

In order to derive a river mask, Elmi et al. (2016) estimate a Maximum A Posterior solution of a Markov Random Field (MAP-MRF), in which the spatial interactions between pixels and temporal variation of the pixel values is considered. Figure





8 (top panel) presents the block diagram of the proposed method. First, the cloud-covered images are removed. Initial water masks are then generated by applying a dynamic threshold. The procedure continues by developing the joint conditional

models, rearranging the problem as one of energy minimization, and developing a graph. In the next step the MAP solution is found by applying the graph cuts technique. Using the MAP solution, the initial water masks and the frequency coverage map are updated, allowing for modifying the already developed graph. The final river mask is then obtained by finding the MAP solution for the modified graph. The uncertainty of the derived masks are estimated by marginalizing the final residual graph ((Elmi et al., 2016; Elmi, 2019)).

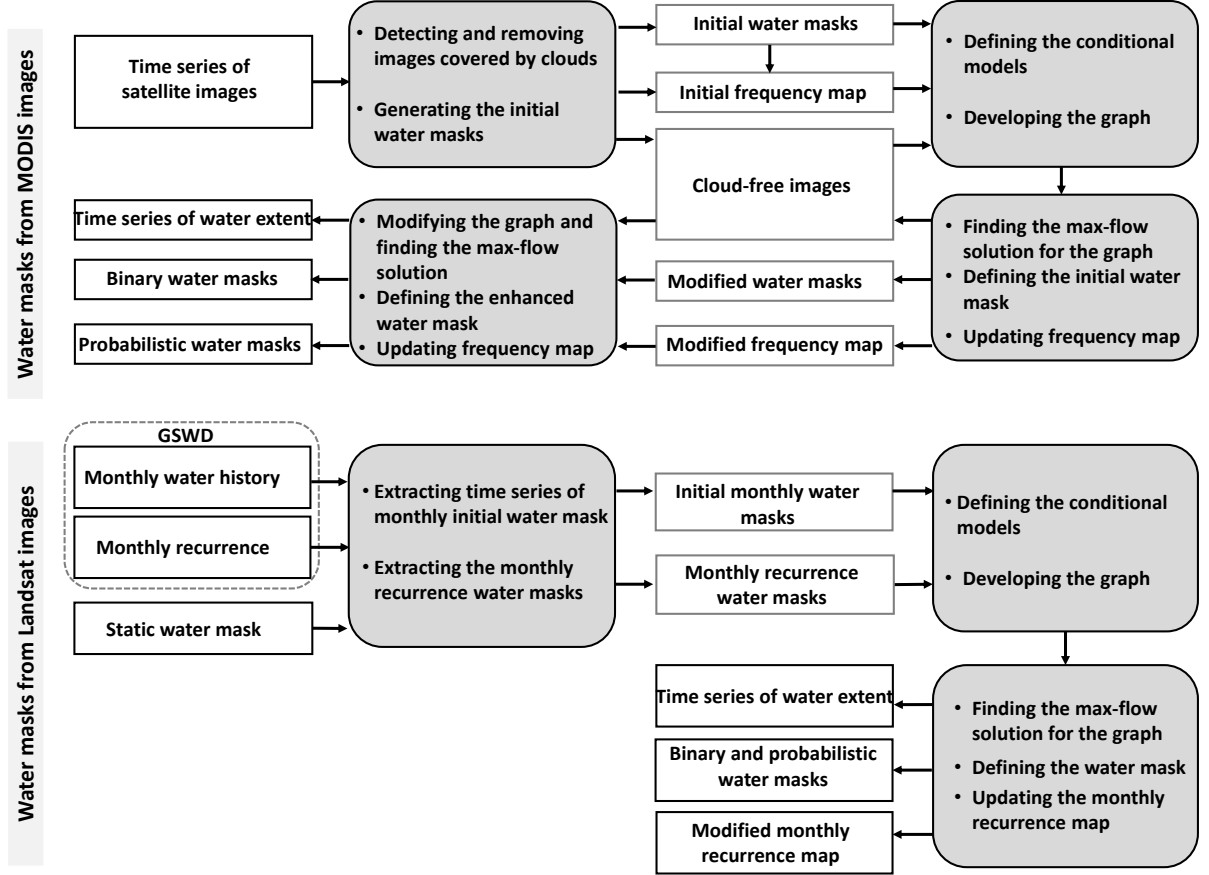

**Figure 8.** Flowchart of the proposed method for generating time series of surface water extent from MODIS images (top pannel). Flowchart of the proposed algorithm for generating time series of surface water extent from Landsat-based GSWD product (bottom panel)

Although MODIS provides homogeneous daily snapshots of the Earth's surface for more than 20 years, its coarse spatial resolution is a limiting factor for generating dynamic river masks of relatively narrow river reaches and small lakes. To tackle this limitation, we use the Global Surface Water Dataset (GSWD) – developed by the European Commission's Joint Research





Centre in the framework of the Copernicus Programm (Pekel et al., 2016) – as an alternative to MODIS data. Derived from Landsat data, GSWD is a unique product for analyzing the spatial and temporal distribution of water surfaces at the global

scale over the past 3 decades. In its estimate of the surface water area, however, the dataset is subject to significant over - and underestimations. The main reason for underestimating the surface water area is the pixels labeled as *No observations* in the dataset. In the GSWD's Monthly Water History product, some pixels are contaminated due to the Scan Line Corrector (SLC) failure of Landsat 7 and cloud coverage. Discarding all pixels with the *No observations* not only leads to an underestimation of the area, but also to the generation of an erroneous water mask. On the other hand, a high noise level in the Landsat images

might be the main reason for a possible overestimation of surface water extent. To generate enhanced dynamic water masks from this dataset, the algorithm described in Figure 8 (bottom panel) is developed (Elmi et al., 2021a). The algorithm performs similar steps as for MODIS relying on the GSWD masks instead of the original images.

    Using the described algorithms, HydroSat provides surface water extent time series of lakes, reservoirs and river reaches from optical satellite images. From the obtained river reach area an effective river width can be determined by dividing the area

by the length of the river reach. Figure 9 shows time series of surface water extent for rivers and lakes derived from Landsat imagery in the Mississippi River Basin.



**Figure 9.** Time series of river reach area of selected river reaches in the Mississippi River basin are in the middle panel.



To analyze the performance of the algorithm, we compare the time series of monthly river masks from 9 river sections (average length of 10 km) in the Mississippi River basin with in situ discharge measurements from nearby USGS stream gauging stations. In all river reaches, we observe a relatively high rank correlation. However, in some river sections, as in

Figure 9(g, f), the sections are too narrow (about 25 and 40 m), even for Landsat imagery with 30 m spatial resolution.

For lakes and reservoirs (second set of time series in Figure 9), the time series of surface water area are compared with altimetric water level time series. For this comparison, 9 small to medium-sized lakes or reservoirs in the Mississippi River basin are selected. In general, water level time series and surface water area show a good agreement represented by the reported rank correlation coefficients. For water bodies with smaller size, it is expected that the rank correlation decreases. As shown

in Figure 9, the rank correlation falls off to 0.57 and 0.61 for Rathbun and Barren Lakes with average areas below $30\,\mathrm{km}^2$ and $50\,\mathrm{km}^2$.

## 4 Water storage anomaly

### 4.1 Water storage anomaly from GRACE and GRACE Follow On

Global observation of total water storage change is vital for understanding the water cycle and climate system dynamics.

The variations of water storage indirectly reflect the Earth's energy storage, ocean heat content, land surface water storage, biogeochemical, and ice-sheet response to global warming (Tapley et al., 2019; Famiglietti, 2004). Water storage variation, both globally and regionally, influences our societies as it affects agricultural, industrial, and domestic water use. Nevertheless, for a long time, monitoring of water storage changes relied on insufficient site measurements, which was costly and time-consuming. Moreover, on continental scales, it was not possible to map water storage because of the sparseness of the station

networks (Rodell and Famiglietti, 1999). Furthermore, measurements of water storage changes by gauging groundwater level and soil water saturation changes are not reliable due to the lack of accurate storage coefficients (Strassberg et al., 2007; Riegger et al., 2012). Hydrological and land surface models have alleviated the problem to some extent. Such models estimate TWS and its components via simplifying real-world systems. The model outputs, however, are subject to high uncertainty and low accuracy due to the lack of global and systematic hydrological data (Jiang et al., 2014).

The GRACE launch in 2002 (Tapley et al., 2004) added unprecedented observations to the existing earth monitoring system. The satellite program was jointly developed by the National Aeronautics and Space Administration (NASA) of the United States and the German Aerospace Center (DLR), which allowed for the recovery of the time variable gravity field at catchment scales using the so-called low-low satellite to satellite tracking concept. The GRACE mission ended in October 2017 while providing over 15 years of near-continuous measurements of the Earth's gravity field. The monthly gravity variations are

used to track mass changes in the hydrosphere, cryosphere, and oceans, quantifying Total Water Storage Anomaly (TWSA). GRACE observations paved the way to monitor continental water storage, including deep soil water, for the first time. It contributed to various applications, including determining the natural and anthropogenic footprints in the global and regional water changes, ice-sheet mass balance (Chen et al., 2009), ocean circulation, sea-level rise (Chen et al., 2013a; Jacob et al., 2012), atmospheric circulation patterns, drought monitoring (Long et al., 2013; Thomas et al., 2014), and flood forecasting





(Reager et al., 2014). GRACE Follow-On (GRACE-FO), launched in May 2018, is continuing GRACE's legacy of monitoring Earth's temporal gravity field using the same constellation as GRACE while additionally equipped with an experimental Laser Ranging Interferometer (LRI).

The collected GRACE/GRACE-FO measurements of each month are processed to estimate the Earth's gravity field in terms of spherical harmonics. The Science Data System (SDS), a joint US/German cooperation consisting of the Jet Propulsion Lab-

oratory (JPL), the University of Texas Center for Space Research (UT-CSR), and the German Research Centre for Geosciences (GFZ), provides quality-controlled data from Level-0 (KBR range data) to Level-3 (grids). Moreover, Level-3 data from the mascons approach in terms of TWSA can be accessed from UT-CSR, JPL, and NASA-GSFC, while the latter does not provide GRACE-FO observations. Other than Level-3 data, several centers help to visualize GRACE TWSA. The JPL and GSFC mascons, for instance, can be visualized using the Mascon Visualization Tool from the University of Colorado Boulder, and

the basin-wise variability of TWS can be obtained from the Gravity over basins Information Service (GravIS) website. Furthermore, several data browsers allow the interactive retrieval of GRACE and GRACE-FO data, including the one developed within the International Center for Global Earth Models (ICGEM) project, the GRACE Plotter, and NASA data Analysis Tool. Table 4 lists the above mentioned centers and products, including HydroSat.

**Table 4.** List of centers which provide Level-3 TWSA from GRACE and GRACE-FO.

| Product | Sensor(s) | Source | Remark |
|---|---|---|---|
| **Level-3 datasets** | | | |
| JPL | GRACE | Landerer (2020e); Landerer and Swenson (2012) | gridded (1 ° spatial resolution), up to d/o 60 |
| | GRACE-FO | Landerer (2020d) | gridded (1 ° spatial resolution), up to d/o 60 |
| | GRACE/GRACE-FO | D. N. Wiese (2020) | mascon approach based on the Level-1 data |
| CSR | GRACE | Landerer (2020b) | gridded (1 ° spatial resolution), up to d/o 60 |
| | GRACE-FO | Landerer (2020a) | gridded (1 ° spatial resolution), up to d/o 60 |
| | GRACE/GRACE-FO | Save (2020); Save et al. (2016) | mascon approach based on the Level-1 data |
| GFZ | GRACE | Landerer (2020c) | gridded (1 ° spatial resolution), up to d/o 60 |
| | GRACE-FO | Landerer (2020a) | gridded (1 ° spatial resolution), up to d/o 60 |
| | GRACE/GRACE-FO | Boergens et al. (2020), http://gravis.gfz-potsdam.de/land | basin-wise |
| HydroSat | GRACE/GRACE-FO | http://hydrosat.gis.uni-stuttgart.de | corrected for leakage and tidal aliasing error |
| **Visualization centers** | | | |
| University of Colorado Boulder | GRACE/GRACE-FO | https://ccar.colorado.edu/grace | JPL and GSFC mascons |
| ICGEM | GRACE/GRACE-FO | http://icgem.gfz-potsdam.de/home | JPL and CSR Level-3 |
| The GRACE Plotter | GRACE/GRACE-FO | http://thegraceplotter.com | Level-3 from various level-2 products |
| NASA | GRACE/GRACE-FO | https://grace.jpl.nasa.gov/data-analysis-tool | Level-3 product from JPL and CSR solutions |

Figure 10 presents the scheme of the data processing handled in HydroSat to retrieve TWSA from GRACE Level-2 solutions.

The ITSG-Grace2018 unconstrained gravity field model from the Institute of Geodesy at the Graz University of Technology (Mayer-Gürr et al., 2018; Kvas et al., 2019) is used as the input to the HydroSat processing. Each monthly solution contains




the full hydrological, cryospheric, and Glacial Isostatic Adjustment (GIA) signal in the form of fully normalized Spherical Harmonic (SH) coefficients, after removing the contributions from other phenomena like tides (ocean, solid earth, and atmospheric), atmospheric and non-tidal oceanic mass changes.

To obtain TWSA, HydroSat applies several corrections on GRACE solutions, known as post-processing steps. The degree-1 coefficients are added to the GRACE solutions, accounting for the movement of the Earth's center of mass (Swenson et al., 2007). Since GRACE estimations of the lowest-degree zonal harmonic coefficient are not accurate, we replace GRACE $C_{2,0}$ and $C_{3,0}$ by the coefficients derived from Satellite Laser Ranging (SLR) data Cheng et al. (2013). The real shape of the Earth is much closer to an ellipsoid than a sphere. Therefore, each solution is corrected from spherical to ellipsoidal coefficients

following the method proposed by Li et al. (2017). To calculate the geoid anomalies, we remove the long-term (2004–2010) mean of spherical harmonics. Due to imperfect tidal models, GRACE SHs are contaminated by residual tidal aliasing error, a primary and a secondary one (Tourian, 2013). Therefore, HydroSat eliminates the primary and secondary tidal aliasing errors of the main tidal constituents, S1, S2, P1, K1, K2, M2, O2, O1, and Q1 from GRACE monthly solutions using a least-squares Fourier analysis (Tourian, 2013).

Furthermore, the monthly solutions are contaminated by noise from different sources including the high-frequency noise in the spherical harmonic coefficients due to the orbit geometry and sensor noise. In order to reduce the high-frequency noise and retrieve mass changes, Gaussian filtering with a radius of 400 km is applied (Wahr et al., 1998). Moreover, to correct for leakage due to filtering, existing methodologies are employed e.g data-driven approach proposed by Vishwakarma et al. (2017) and forward-modelling (Chen et al., 2013b). Finally, HydroSat corrects the GIA signal following (A et al., 2012). The

uncertainty for the GRACE-based TWSA at each month is estimated by propagating the calibrated error of the GRACE and GRACE-FO level-2 solutions.

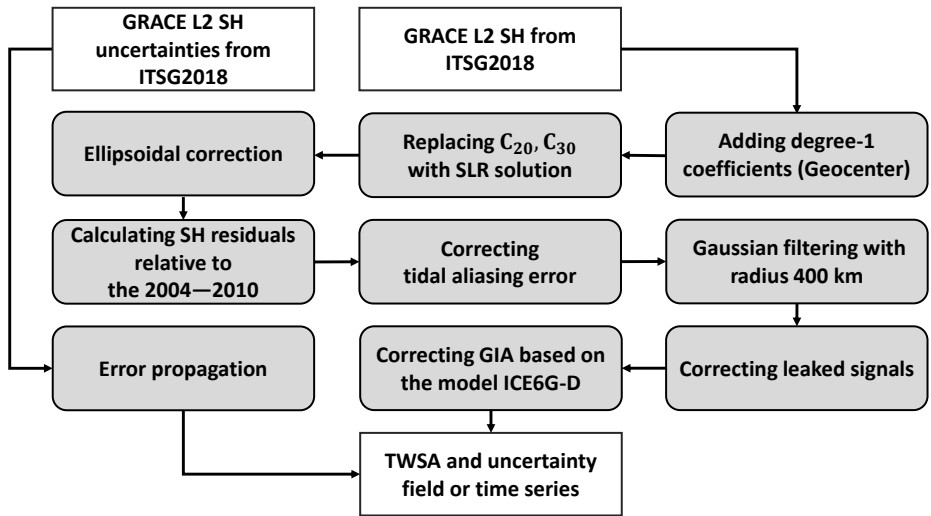

**Figure 10.** Flowchart of obtaining Total Water Storage Anomaly (TWSA) in HydroSat



Figure 11 (top) shows the long-term trend of the global major hydrological basins. Figure 11 (bottom) compares the time series of GRACE and GRACE-FO TWSA estimation over the selected basin from HydroSat (black) with two mascon products: CSRv2 (red) and JPLv2 (blue). For visual simplicity and better comparison, only two common and updated mascon products

have been selected. In general, in all catchments, TWSA estimation follows the two mascons products well. Minor discrepancies are observed over Yangtze river basin and Murray-Darling, which can be explained due to the different GIA models and low signal to noise ratio, respectively.



**Figure 11.** Top: Annual trend of TWSA over global major river basins from april 2002 to 2020. Selected basins are numbered in the map. The borders are defined based on GRDCs major river basins of the world (https://www.bafg.de/GRDC). Bottom: Time series of TWSA from GRACE and GRACE-FO for the selected basins, comparing the results from HydroSat, JPLv02 global mascons, and CSRv02 global mascons. The corresponding HydroSat number for each basin is provided in the gray box.





## 4.2 Lake and reservoir water storage anomaly

The dynamics of lake and reservoir storage is a key parameter in studies about the global hydrological cycle. The variation
of lakes and reservoirs water height and surface area have been successfully monitored using spaceborne measurements, as
demonstrated in previous studies. Table 5 lists some of these studies providing either time series of water volume change or
lake and reservoir bathymetry.

**Table 5.** Overview of studies proving time series of lakes and reservoirs water volume variations

| Study | Water level | Surface area | Remark |
|---|---|---|---|
| Busker et al. (2019) | DAHITI | GSWD | water volume variation of 137 lakes are provided |
| Crétaux et al. (2011) | HYDROWEB | MODIS, Aster, Landsat, Cbers | water volume variation for 100 lakes are provided in Hydroweb |
| Schwatke et al. (2020) | DAHITI | DAHITI | water volume variation of 62 lakes and reservoirs are available on 2021 |
| Klein et al. (2021) | DAHITI | Global WaterPack | water volume variations of 1267 reservoirs are analyzed the dataset is not publicly available |
| Li et al. (2020) | ICESat, Hydroweb, G-REALM | GSWD | 3-D reservoir bathymetry of 347 global reservoirs are provided |
| Gao et al. (2012) | HYDROWEB | MODIS | water volume variation of 34 reservoirs from 2000–2010 |
| Li et al. (2019) | Jason 1-3, ENVISAT, Cryosat, ICESat | Landsat, GaoFen-2 | water volume variation of 52 large lakes on the Tibetan Plateau during 2000–2017 |

The mentioned studies follow almost the same strategy to generate water volume anomaly time series or a bathymetry
map. In these studies, after collecting the simultaneous surface water area and level measurements, the empirical relationship
between lake water level and area is developed. Then the water volume variations are estimated using by the so-called *end-area
formula* or pyramid formula.

Similar to previous studies, HydroSat estimates the water volume anomaly for lakes and reservoirs following a straightfor-
ward approach relying on monotonic relationship of water level and surface area (Figure 12). The algorithm starts by acquiring
the water area from satellite imagery (explained in Section 3) and the time series of the water level (explained in Section 2).
To derive the time series of the water volume anomaly and also the water level-area-volume model, the following steps are
performed:

   – Generating monthly water level time series with the same temporal sampling as the surface area time series

   – Creating the scatterplot of the simultaneous surface water area and level and removing the blunders in the scatterplot

   – Defining the water surface area-level model through either parametric and non-parametric approaches





– Estimating the water volume variations by assuming that between two successive pair of measurements (water level $H$ and lake area $S$), the lake morphology is regular and has a pyramidal shape, (Abileah et al., 2011).

$$\Delta V = \frac{\left[H(t) - H(t-1)\right]\left[S(t) + S(t-1) + \sqrt{S(t)S(t-1)}\right]}{3} \tag{1}$$

– Obtaining the water area-level-volume model

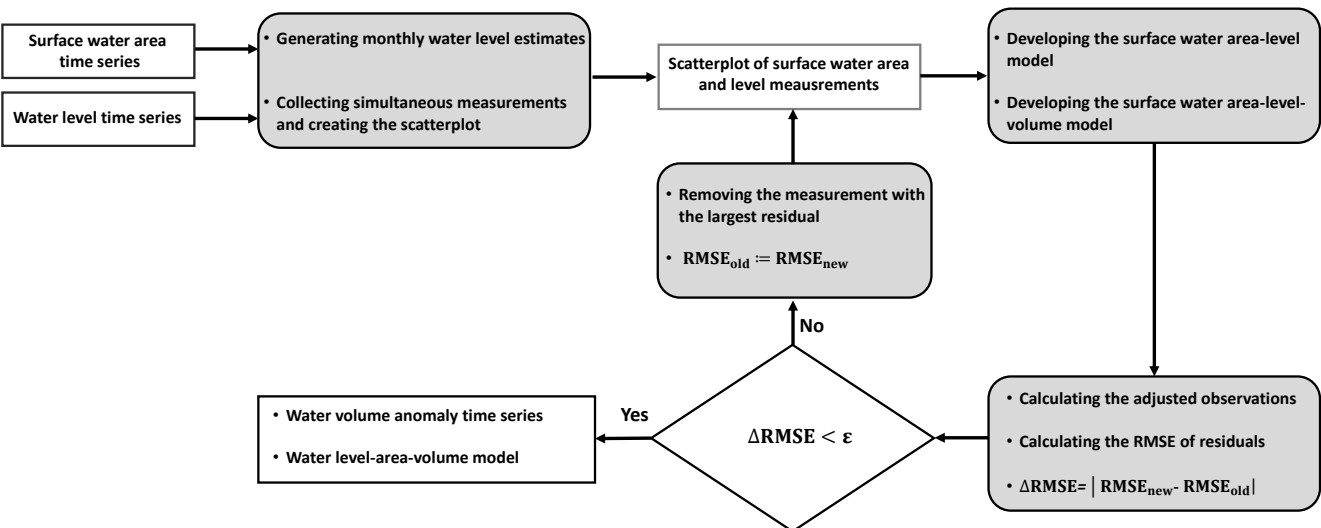

**Figure 12.** Flowchart of obtaining water volume anomaly using surface water area from satellite imagery and water level time series from satellite altimetry

Figure 13 shows time series of water volume anomaly for some small, medium and large lakes with different climate
characteristics. A statistically representative joint time period for altimetry and imagery data is crucial to obtain a reliable estimate of water volume. For example Tana, Sobradinho, and Dale Hollow Lakes, simultaneous surface water area and water level measurements are available for more than 10 years, resulting in a reliable area-volume model. Over the Arkabutla Lake and Barren Lake, the joint time period is rather short but representative as it covers the entire statistical distribution of both variables. On the other hand, the time series of lake volume anomaly of McConaughy, Harlan County, Rathbun and Mark
Twain lakes carry mismodelling error as their joint time periods are short and non-representative of the entire distribution.

## 5   River discharge from space

Monitoring of river discharge, defined as the volume of water passing a river section in a given time, is a critically important part to understanding a broad range of science questions focused on hydrology, hydraulics, biogeochemistry and water resources management. Especially the river discharge quantification in ungauged basins anywhere and anytime is the holy





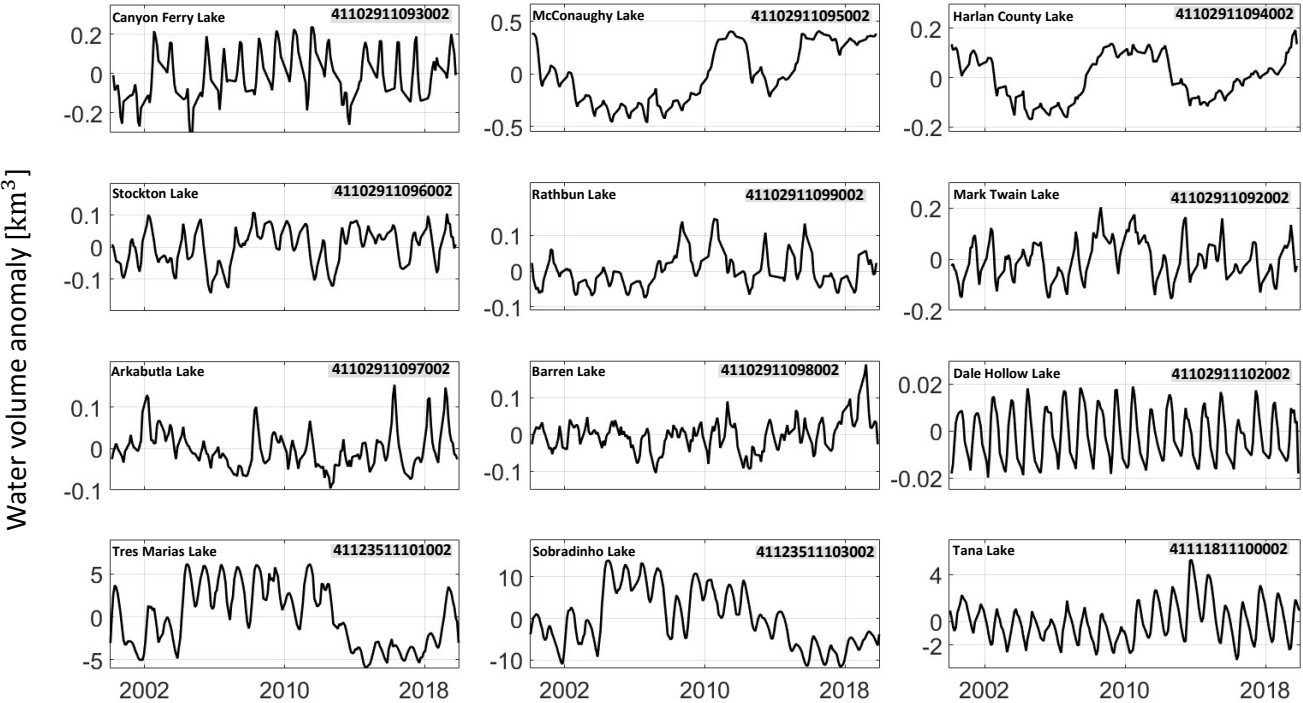

**Figure 13.** Time series of water volume change of selected lakes and reservoirs. The location of these lakes can be found in figures 4 and 9

grail of hydrology. River discharge reflects the drainage basin dynamics and affects environmental conditions like currents and hydrography in coastal waters; it is a function of precipitation and meteorological elements controlling evapotranspiration, geology, relief, and vegetation.

Despite its importance, the publicly available in situ river discharge database has been declining steadily over the past decades due mainly to economic and political reasons. From about 8000 (pre-1970), the number of available gauging stations has decreased to less than 1000 (around the year 2015) (Lorenz et al., 2014, 2015; Tourian et al., 2017b).

Given the insufficient monitoring from in situ gauge networks, and without any outlook for improvement, spaceborne approaches come to the rescue. Satellite-based Earth observation with its global coverage has been demonstrated to be a potential alternative to in situ measurements. In future, the SWOT mission with wide-swath altimetry is expected to attain global river discharge given its unprecedented temporal resolution, spatial coverage and the synchronous availability of river height, width and slope (Biancamaria et al., 2016; Durand et al., 2016).

In HydroSat, discharge estimates are available from both altimetric river water level and imagery-based effective river width also in the two modes of Standard-Rate (SR) and High-Rate (HR). While the SR product relies on standard temporal resolution of the spaceborne data, the HR data comes with a higher temporal resolution through an assimilation process. To the best of our knowledge, there is no repository or website providing similar space-based river discharge estimates.





## 5.1 Standard-Rate river discharge

Standard-Rate river discharge $Q$ at selected gauges can be estimated from space-based water level $H$ (or river width $W$) measurements using an empirical relationship between river height (or width) and discharge (Kouraev et al., 2004; Birkinshaw et al., 2010). The most common form of this relationship is the so-called *rating curve* $Q = F(H)$ or $Q = F(W)$. Rating curves are conventionally generated for simultaneous measurements of space-based water level (or river width) and in situ river discharge. Once the model is developed, the discharge can be determined from water level or width measurements. The restriction however remains that for deriving a rating curve, simultaneous measurements are required, meaning the availability of in situ measurements during the satellite era.

Globally a great portion of existing gauges are not active during the satellite era although they provide a wealth of legacy data. For such gauges, Tourian et al. (2013) suggest a statistical approach based on quantile mapping of in situ discharge and altimetry water level measurements. Since the quantile functions of discharge and river water level (width) have a same *x*-axis (cumulative probability), it is possible to connect their *y*-axis directly and obtain $F(.)$. As this approach does not involve the time coordinate explicitly, the requirement for synchronous datasets is obsolete. This is to say that the pre-satellite river discharge data records can be salvaged and turned into usable data for the satellite altimetry or imagery time frame.

The method is further improved by Elmi et al. (2021b) to infer a nonparametric model for estimating the river discharge and its uncertainty. The algorithm employs a stochastic quantile mapping function scheme by iteratively 1) generating realizations of river discharge and height (width) time series using a Monte Carlo simulation, 2) obtaining a collection of quantile mapping functions by matching all possible permutations of simulated river discharge and height (width) quantile functions, and 3) adjusting the measurement uncertainties according to the point cloud scatter. The flowchart in Figure 14 describes the procedure of SR discharge estimation using spaceborne river height or width.



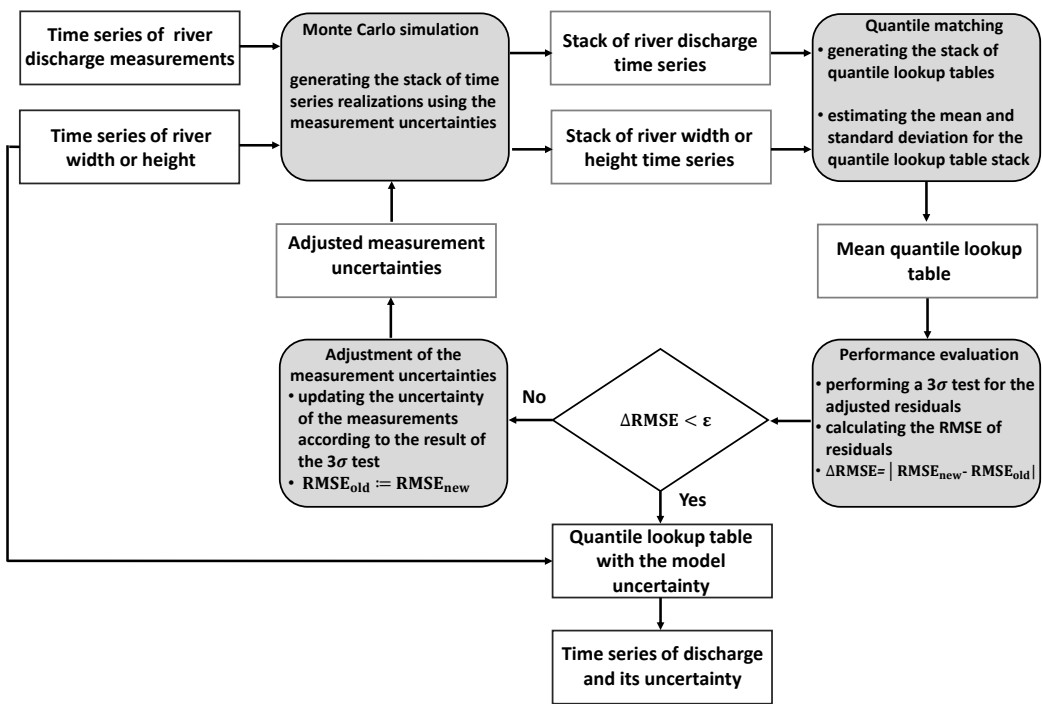

**Figure 14.** Adopted from Elmi et al. (2021b): Flowchart of discharge estimation algorithm for generating the quantile mapping function

Figure 15 presents the discharge estimated from river width, using the stochastic quantile mapping function algorithm over four different river reaches along the Niger, Congo and Po rivers (Elmi et al., 2021b). High NSE values for the Niger River reaches (Figure 15 (a,b)) shows that the developed method can accurately estimate the discharge given that both discharge and width measurements have a representative statistical distribution in the training period. The performance of the method significantly decreases over the Congo River reach (Figure 15 (c)) mainly because of the complex relationship between river width

and discharge in this part of the river. The performance of the algorithm over the Po River reach is only minimally acceptable (NSE=0.13). Here, the discharge is not estimated accurately due to insufficient width measurements at high discharge and low signal-to-noise ratio of the same measurements. The obtained rating curve over the Po River, however, highlights the advantage of using a non-parametric model through the quantile mapping function compared to choosing a parametric model. With the help of the proposed method, spaceborne discharge estimates can be obtained for all non-active gauges in politically-ungauged

basins (see Gleason and Durand (2020)).

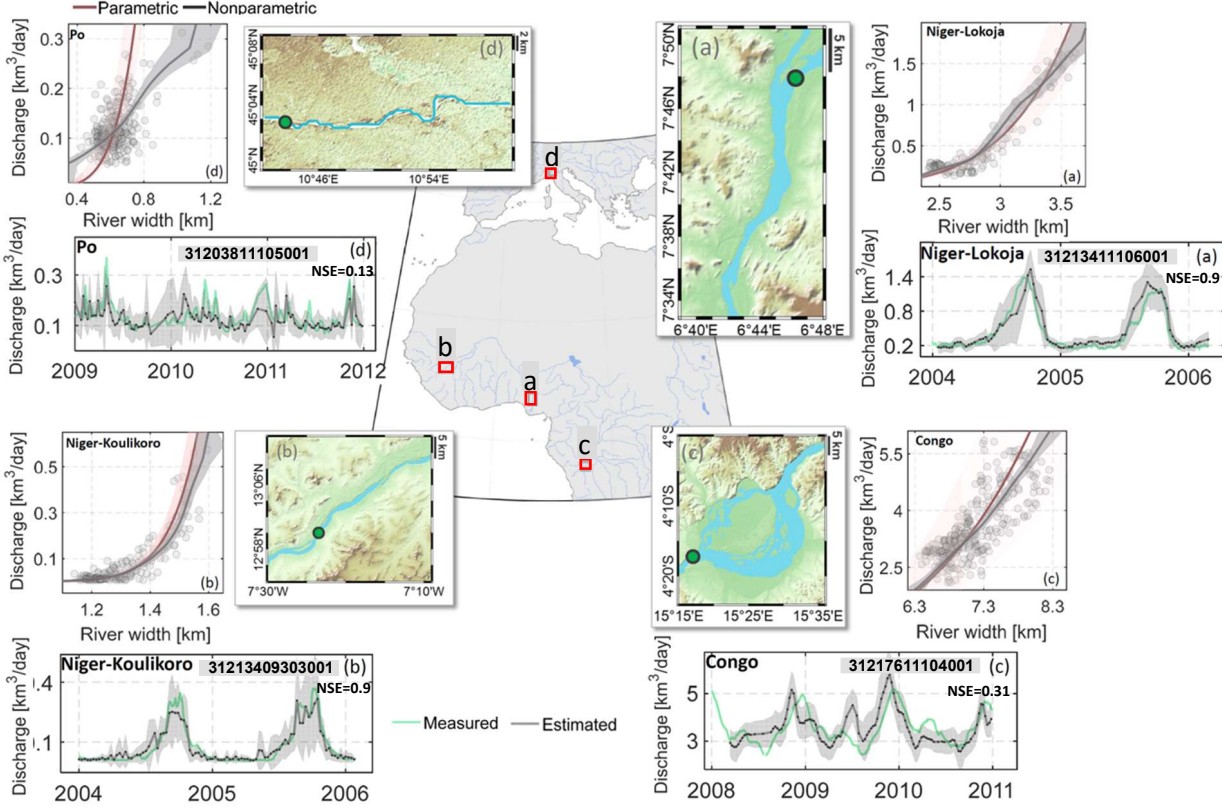

**Figure 15.** Adopted from Elmi et al. (2021b): Four river reaches defined as case studies (a) and (b) are the river reaches over the Niger River. (c) is part of Congo River and (d) is selected over the Po River. For each case, the estimated discharge (green dots), scatterplots of the simultaneous observations used for developing models together with the developed rating curve models and stochastic quantile mapping function.

## 5.2   High-Rate river discharge

HydroSat provides High-Rate (HR) river discharge time series based on the method developed by Tourian et al. (2017a) that goes beyond the conventional one-on-one relationship between virtual station (or reach) and (legacy) in situ station explored in SR discharge products. A multitude of altimetric discharge time series over a river network are used in this approach to estimate time series of daily river discharge. This is fundamentally done via assimilating multiple altimetric discharge – the SR time series – and in situ measurements using a linear dynamic system. The dynamic system consists of a stochastic process model that benefits from the cyclostationarity of discharge. This model is informed by the covariance and cross-covariance generated out of old in situ data. The process model is then combined with observation equations fed by several altimetric and in situ discharge time series to form a linear dynamic system. Ultimately, the system is solved using the Kalman filter, followed by smoothing the solutions using the Rauch-Tung-Striebel (RTS) scheme (Rauch et al., 1965). In fact, the Kalman







filter produces an a posteriori discharge estimate with a likelihood function of discharge based on the available observations and the prior information derived from the stochastic process model. In case of an observation gap, the posterior estimates rely on the stochastic process model and its cyclostationary mean discharge. Figure 16 represents the flowchart for estimating the HR river discharge. The implementation details of the method can be found in Tourian et al. (2017a).

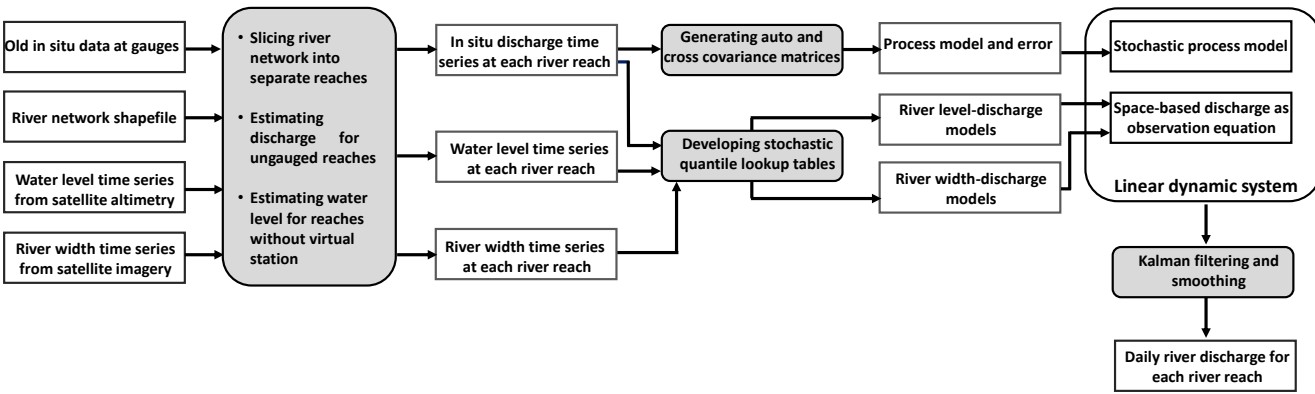

**Figure 16.** Flowchart of estimation of High-Rate river discharge using a Kalman filter approach

Figure 17 shows an example of a HR river discharge time series over River Niger at Lokoja. The inputs to the dynamic system are the altimeric and in situ river discharges. During the period when the only source of observations is the in situ data (before 1992), the Kalman filter estimates a discharge that matches the in situ data with a relative RMSE of less than 2% (not visible in the figure). After 1992, when altimetric river discharge is available, the Kalman filter is less dependent on the in situ data, leading to a relative RMSE up to 50%. The validation over the entire Niger basin and 22 gauges along the main stem 480   show an average correlation of 0.9, an average relative RMSE, a relative bias of about 15%, and a NSE greater than 0.5 for 15 gauges.

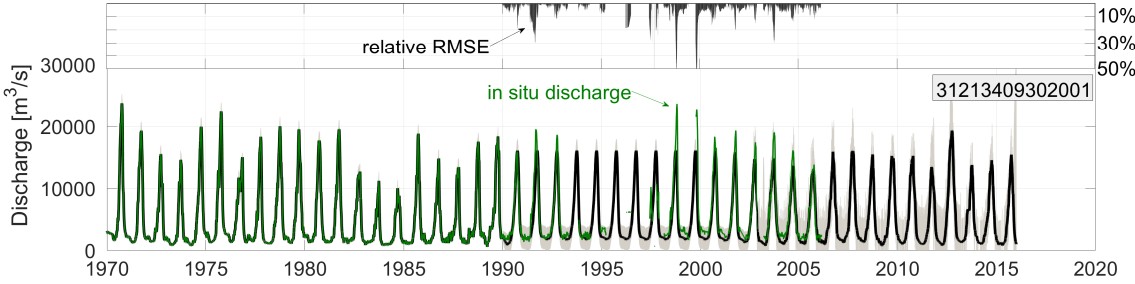

**Figure 17.** Daily river discharge over the Niger River at Lokoja by assimilating altimetric discharge with available in situ data

    The proposed method is applicable in all river networks with available legacy in situ data. It allows for obtaining a smoothed daily discharge time series without data outages at any given location along a river.



## 6 Summary, conclusion and outlook

The development of repositories and services to provide global water cycle products from spaceborne sensors is getting more attention than ever before, which is motivated by the urgent need for more hydrological evidence, the absence of perspective for improving in situ data, the existence of an abundance of satellite missions, the prospect of cutting-edge missions such as the SWOT mission, and also the promise of operational satellites in space. Such products support studies focused on understanding the water cycle and the Earth system in general. HydroSat as a repository of global water cycle products provides estimates and uncertainty of the following water cycle variables:

- **surface water extent of lakes and rivers**: HydroSat provides surface water extent time series of both lakes and rivers from optical satellite images. For generating dynamic water masks, region-based classification is employed, which benefits from the spatio-temporal behavior of pixel-intensity. This allows us to deal with the complexities in extracting dynamic river masks. Moreover, such an algorithm setup allows obtaining a probabilistic water mask leading to an estimate for surface water extent uncertainty. While datasets of surface water extent variation over lakes are available from various data centers, HydroSat additionally provides time series of river width for major river basin. For a quality assessment, time series of surface water extent over lakes are compared with available in situ and or altimetric water level time series. Over rivers, such a quality assessment is predominantly done through comparing the time series against in situ river discharge.

- **water level time series of lakes and river**: HydroSat provides water level time series of rivers, lakes and reservoirs in two modes, standard-Rate (SR) and High-Rate (HR), with their uncertainty estimates. For water level time series HydroSat uses an automated, data-driven outlier rejection methodology designed within an iterative, non-parametric adjustment scheme. The outlier-free measurements form the final time series without any further smoothing. While water level time series over inland water bodies are available from similar data centers, HydroSat additionally provides HR water level time series over rivers through densifying individual SR time series along a river. For the HR products over the lakes, inter-satellite biases are removed through a hybrid approach by incorporating lake surface area information. The quality of water level time series is assessed through a validation against in situ water level or proxy data like river discharge, river width or lake surface area.

- **terrestrial water storage anomaly**: HydroSat provides Terrestrial Water Storage Anomaly (TWSA) time series and its uncertainty over global major river basins using GRACE and GRACE-FO observations. To estimate TWSA time series from level-2 data (shperical harmonics up to degree/order 96), the $C_{2,0}$ and $C_{3,0}$ are replaced and degree-1 is added from the corresponding SLR estimates. Moreover, ellipsoidal and GIA correction are followed together with a smoothing Gaussian filter with a radius of 400 km. The final TWSA in HydroSat are corrected for tidal aliasing error and leaked signals. For the field product, the Gaussian filtering is applied together with a de-striping filter and leakage is corrected using forward modeling approach. The quality of TWSA time series is assessed through a comparison with two mascons products, CSR RL06 version 02 and JPL RL06 version 02.





– **water storage anomaly of lakes and reservoirs**: HydroSat provides time series of surface water storage anomaly for lakes and reservoirs using the time series of water level and surface area measurements. For developing the surface water level-area-volume model, the scatterplot of simultaneous water level and area measurements is obtained. HydroSat

performs an iterative data snooping procedure to obtain a reliable empirical relationship between surface water level and area. In this way, the quality of the obtained time series of water storage anomaly is ensured, since the non-representative measurements are eliminated.

– **river discharge estimates for large and small rivers**: HydroSat provides SR and HR river discharge estimates together with their uncertainties. To obtain the SR products, HydroSat relies on a non-parametric quantile mapping approach that

salvages gauging stations that are no longer updated with in situ measurements. Since no model assumptions are required under the non-parametric approach, the HydroSat discharge time series are less contaminated by a mis-modeling. For the uncertainty estimation HydroSat applies a stochastic quantile mapping function algorithm supported by a Monte Carlo simulation. The availability of enough SR time series over a river network allows approximating the spatio-temporal dynamics of a river system by a linear dynamical system. The HR products are the solutions of such a dynamic system

by Kalman filters obtained with up to daily temporal resolution at potentially any location along the river. For the quality assessment, SR and HR discharge time series are compared with in situ and spatial river discharges, water levels, and river widths.

The above hydrological variables can directly be use in hydrological modelling, either as inputs or for calibration purposes. Global hydrological modeling has been improved in terms of modeled processes leading to a better recognition of modeling

uncertainties. Such recognition clearly signals that the modeling uncertainties are not reduced. To reduce the uncertainty one solution is indeed, to use the best available input data for modelling. Space-based data together with process knowledge allow a realistic modelling of water flows and storages in the different compartments. Moreover, spaceborne measurements can be used to calculate indicators of the past and future state of the global freshwater system to assess risks under $1.5°C$ and $2°C$ global warming and support decision making (Döll et al., 2018).

In addition, variables of water level, surface water storage anomaly and surface water extent support downscaling of mass transport monitoring in time and space. The success of GRACE has created a new demand for scientists and decision makers for a sustained observation of the terrestrial water storage change (Pail et al., 2015). Although the utility of GRACE data has been mainly limited to large catchments, understanding of water storage changes in regions with some local weak signatures play an important role within the Earth system and the sustainable development of water resources (Lorenz et al., 2015). Therefore, an

urgent priority is to mitigate the spatial resolution limitation of GRACE through incorporating additional hydrological variables such as surface water storage and river and lake water level variation. This will improve in particular our understanding of the water cycle in many small vulnerable catchment areas with large populations.

Another added value of the HydroSat data is its complementary role to the SWOT mission. Over rivers, SWOT will estimate discharge from multiple algorithms as well as consensus values computed over multiple individual algorithms (Stuurman and

Pottier, 2020). The majority of algorithms are Bayesian, relying on prior data. Hydrologic variables provided by HydroSat





can effectively be used as potential prior information for each of the discharge algorithms through available water levels (SR and HR), river width from satellite imagery, and discharge estimates (SR and HR). Over lakes and reservoirs, our estimates of surface water extent and volume anomaly will boost SWOT's estimates both in terms of temporal resolution and coverage. This supports studies aiming to understand long-term behavior of lakes and reservoirs.

## 7 Data availability

The data are publicly available in the HydroSat repository via http://hydrosat.gis.uni-stuttgart.de. All time series in this paper are assigned a number, the so-called HydroSat ID, by which the time series can be found in the HydroSat repository via the search field. In this repository all data can be browsed, visualized and analyzed without registration. However, registration is required to download the data. A snapshot of all the data (taken in April 2021) is available in GFZ Data
Services, which is accessible during the peer review process via https://dataservices.gfz-potsdam.de/panmetaworks/review/ a250099b6bb14c162399cf78b1a3182b1e4420b2db8373ebd6a647d50c0e0326/ (see *Files*) and will be published at https://doi. org/10.5880/fidgeo.2021.017 (Tourian et al., 2021).

*Author contributions.* MT conducted the project, wrote the majority of the paper and together with OE and YS devised HydroSat, defined requirements, designed repositories layout and features. OE wrote sections 3, 4.2 and contributed in Section 5.1. YS developed the HydroSat
website during his time at the University of Stuttgart. SB wrote sections 2.1 and 2.2.1. PS wrote Section 4.1. RS maintains the HydroSat website. NS reviewed the paper and contributed in HydroSat activities. All authors discussed the results, reviewed the manuscript and commented on it.

*Competing interests.* The authors declare that there is no conflict of interests regarding the publication of this paper.

*Acknowledgements.* The authors acknowledge Dr. Alireza Sahami, Apple Inc., for his valuable inputs within the early phase of HydroSat.
The authors also like to thank Mr. Thomas Götz, Institute of Geodesy, University of Stuttgart, for his support regarding database and server of HydroSat repository. The authors acknowledge following data centers for providing satellite data

– Envisat GDR-v3 data from ftp://ra2-ftp-ds.eo.esa.int https://doi.org/10.5270/EN1-ajb696a

– Saral GDR T data from ftp://ftp-access.aviso.altimetry.fr/geophysical-data-record/

– ICESat2 ATLAS/ICESat-2 L3A Inland Water Surface Height, Version 3 from https://nsidc.org/data/atl13

– CryoSat-2 SIR GDR data from ftp://science-pds.cryosat.esa.int/

– Jason-1 GDR data from ftp://ftp-access.aviso.altimetry.fr/geophysical-data-record/

– Jason-2 (PISTACH) GDR data from ftp://ftpsedr.cls.fr/pub/oceano/pistach/J2/IGDR/hydro/



- Jason-2 GDR data from ftp://ftp-access.aviso.altimetry.fr/geophysical-data-record/

- Jason-3 GDR data from ftp://ftp-access.aviso.altimetry.fr/geophysical-data-record/

- Sentinel-3A NT data from https://scihub.copernicus.eu/dhus/home

- Sentinel-3B NT data from https://scihub.copernicus.eu/dhus/home

- GRACE monthly data ITSG-Grace2018 from https://doi.org/10.5880/ICGEM.2018.003

- GRACE-FO monthly data ITSG-Grace2018 from https://doi.org/10.5880/ICGEM.2018.003

- MODIS MOD09Q1 data from https://ladsweb.modaps.eosdis.nasa.gov/

- Landsat based water masks from https://global-surface-water.appspot.com/



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
