# Peer review of "HydroSat: a repository of global water cycle products from spaceborne geodetic sensors"

_Earth System Science Data, 2021_

## Author Comment (AC1)

**RC1**: 'Comment on essd-2021-174', Anonymous Referee #1

*Manuscript handles a very important topic of creating a database for hydrological studies. Authors combined various satellite datasets with an aim to provide consistent output of surface water extent of lakes and rivers, water level of inland water bodies, water storage anomaly of hydrological basins, lakes and reservoirs, and river discharge for large and small rivers. Paper is original and unique. It also consists of huge science behind the output datasets. Despite its advantages, some shortcomings may be found. These are listed below:*

1. *Paper is very long. It is sometimes hard for the reader to stay focused on the main research idea. I would advise the authors try to shorten the manuscript by putting some less important information into Appendices. It would make the paper more clear.*
2. *There are too many technical details provided within the manuscript. These could be moved, as mentioned above.*

While we were writing we were very much aware of the length of the manuscript and tried to strike a balance between completeness and conciseness. Your comments mean that our sense of balance is not in line with the reader's senes. Therefore, in the revised version, we have moved several flowcharts and figures to the Supplement and shortened the text so that the revised manuscript is 9 pages shorter.

3. *I am not really sure why SWOT mission is mentioned several times across the manuscript. It does not bring any new conclusion or outlook on the results the authors provide. I would advise these comments to be removed.*

Indeed, the driving force behind HydroSat at its inception was not the SWOT mission yet. However, the reason for the frequent references to the SWOT mission in the current manuscript is that SWOT did become over time a sort of guiding framework. The added value of the HydroSat products to the SWOT mission is highly relevant, as it was highlighted in the Summary and Conclusion section. Across the rivers, SWOT estimates discharge using several algorithms as well as consensus values calculated from several individual algorithms. Most of the algorithms use Bayesian principles, i.e., they rely on prior data. Hydrologic variables provided by HydroSat can effectively be used as potential prior information for each of the discharge algorithms through available water levels (SR and HR), river width from satellite imagery, and discharge estimates (SR and HR). In the case of lakes and reservoirs, the time series of surface water extent and volume anomaly provided on Hydrosat will improve SWOT estimates in terms of both temporal resolution and coverage. This supports studies aimed at understanding the long-term behavior of lakes and reservoirs.

In the revised manuscript, we have made the beneficial role of HydroSat for the SWOT mission more explicit in the introduction.

4. *I really miss any comparison of dataset to the existing databases. I mean from the Figures you provide it is clear that your dataset is more or less similar to the existing ones, but maybe some statistics would be useful. At least to show why your dataset is unique and why it should be used instead of others.*

We have intentionally refrained from providing statistics in this manuscript for the following reasons:

- Many databases, including HydroSat, are constantly updating their products. Therefore, statistics based on a particular snapshot of the data could yield a misleading conclusion about a database.
- Due to the lack of in situ data, performing a statistical analysis of all available data from all available databases/repositories is not possible. Therefore, providing statistics on a few selected cases would never be representative and could provisionally downgrade or upgrade a particular database.
- The geographic distribution of virtual stations offered by different data providers is also very different. Such comparison will at the end be highly limited to less complicated case studies.
- We believe that the scientific community needs more HydroSat-type data repositories for a better understanding of the freshwater cycle. Lumping them with statistical metrics would lead the activities to a winner-loser game that is definitely not constructive for further development of similar products.
- Finally, we believe that the uniqueness of the HydroSat global data is legitimised by their availability in many ungauged basins and their scientific basis.

5. *Main advantages of database should be pointed in the introduction section. It is now not really clear where the uniqueness of your database is. While I believe it is new, it should be strictly pointed at the beginning of the manuscript.*

Indeed. In the revised manuscript, we have highlighted the main advantage of the HydroSat in the introduction section. Thank you very much for this comment.

6. *I would advise the authors to clearly divide the manuscript into sections covering introduction, datasets, methodology, results and comparisons, conclusions. For now all the above sections are somehow mixed, which made it hard for me to follow the main topic.*

Considering that we have different products with different input data and methods, a classical structure with Introduction, Datasets, Methodology, Results would probably bewilder the readership. But we do acknowledge your point that the manuscript needs a clearer structure. So instead, the revised manuscript is restructured so that we use the same structure (sub-sections) for each product as described below:

- A short introductory text (with references to other existing datasets)
- HydroSat products
- Data and methodology

This indeed made the manuscript much more structured and clear.

7. *It is not really clear which satellite missions are employed for creating individual outputs. (Or maybe I missed it somewhere?). It would be better if all datasets are strictly listed or mentioned/described.*

This is now clear in the subsection "Data and methodology" for each product. Moreover, in the caption of figures with time series we provided the input data (wherever it was not clear).

---

## Author Comment (AC2)

**RC2**: 'Comment on essd-2021-174',, Anonymous Referee #2

The paper presents a rather unique dataset that provides several hydrological variables obtained with satellite data related to: 1) water level time series of rivers and lakes; 2) surface water extent of rivers and lakes; 3) terrestrial water storage anomaly; 4) water storage anomaly for lakes and reservoirs; 5) river discharge estimates for rivers.

Globally, I found the idea to collect all these variables together quite interesting and useful for hydrological applications even if the dataset is far from to be exhaustive and complete, at least for some variables: for example, water level and water storage anomaly cover quite all the globe, whereas the surface water extent and the river discharge are estimated only for some stations (and sometimes not coincident).

About the paper, it is very long with a lot of information and a few innovative elements. Actually, most (maybe all) of the procedures to derive the hydrological products have been already published. Therefore, the paper shows a collection of already published algorithms and procedures with a remark of the main results and validation. I really appreciated the comparison with other datasets, but I am quite dubious about the general content.

We thank you for your review and sharing your concerns. First of all, the length of the manuscript was also pointed out by another reviewer. So, after transferring some materials to the supplement and restructuring the text, the revised manuscript is shorter and much clearer now.

Regarding the structure of the paper, we would like to point out that we followed the ESSD guidelines for data description papers, in which detailed analysis (as in a research article) remains outside the scope. According to ESSD, articles in the data description category should not focus on instrumentation, methodology, data extraction, or data treatment except when that information helps quantify uncertainties or otherwise facilitates validation of data presented

It is indeed true that there are published papers behind many of these data sets. However, none of these published studies have demonstrated the applicability of their proposed method at a large scale. In the geosciences and remote sensing, applying an existing method to a new case study often brings new scientific challenges that require further modifications. This is exactly the case with most of the HydroSat products. In fact, here the focus is the data itself and the aspect of having unique datasets over many ungauged basins around the world. We describe the methodology to help the readers to understand the caveats and uncertainties involved in the data and also to facilitate comparison and validation of presented data.

In addition, the citation to four unpublished papers (Behnia et al. and Elmi et al.) makes this paper more uncertain for two reasons: first, the content described in the papers is not still accepted by the scientific community, and if from one hand it is good for the originality of the content of this article, on the other hand, no many details are provided on that specific procedures to allow the acceptance; second, no way to check the under review or submitted papers, to verify the originality of the content of this paper.

**RC2**: 'Comment on essd-2021-174',, Anonymous Referee #2

One of these four unpublished works has now been published. In the revised manuscript, we have excluded the references to the unpublished works. The methods are described in the text and in the Supplement.

So, despite it is a big paper with a lot of science behind, I think it is not good enough to demonstrate the novelty of the datasets.

Please see our response to your first point. We see the novelty of the work in the data itself and the fact that we offer a global dataset for water cycle monitoring. Apart from that, as mentioned above, there are modifications to already existing methods described in this manuscript for the first time (see the previous point), which elevates the novelty of the work.

---

## Author Response (AR2)

In this manuscript, Tourian et al. produced a new global dataset on water cycle HydroSat. This dataset was compiled using existing satellite data and their derived products. They also estimate the uncertainty of this product. This observation-based product is potentially useful for understanding the water cycle, improving hydrological models, and assessing freshwater availability.

I think this is an interesting paper and fits into the scope of ESSD. While it has the potential to be published, I have some major concerns on the clarity and novelty and suggest a significant revision on these issues.

The title and abstract are confusing, potentially misleading given the broad readership of ESSD. The water cycle includes several fluxes (Precipitation, ET, surface/subsurface runoff) and stocks (snow, glaciers, lakes and reservoirs, soil moisture, groundwater). The compile dataset only includes estimates for limited components of the water cycle. Why other components are excluded even through relevant satellite observations are available?

Thanks for this comment. Indeed, this is true that our products do not cover the entire water cycle variables. We have changed the title of the manuscript to "HydroSat: geometric quantities of the global water cycle from geodetic satellites". We believe that the term "geometric" characterises all provided variables: water level from satellite altimetry, surface water extent from satellite imagery, terrestrial water storage anomaly in terms of equivalent water height from satellite gravimetry, lake and reservoir water storage anomaly from a combination of satellite altimetry and imagery, and river discharge from either satellite altimetry or imagery.

The spatial and temporal coverage of this dataset should be explicitly mentioned in the abstract.

The spatial and temporal coverage of our products varies and depends on the availability of geodetic satellites. We clarified this point in the abstract.

The novelty of this dataset is unclear to me in the current version. I agree that satellite observations provide a new dimension for understanding the water cycle. Some satellite-based products have been generated. For example, the Hydroweb dataset provides historical and operational water levels for lakes and rivers and GRACE-derived TWS has also been reported in some papers. The authors should highlight why we need a new dataset in the abstract and potentially who are the targeted users in the main text. The improvements of this new dataset upon existing datasets seem to be unclear.

HydroSat is unique since it provides five geometric quantities of the water cycle:  1) water level, 2) surface water extent, 3) terrestrial water storage anomaly, 4) lake and reservoir water storage anomaly, and 5) river discharge from either satellite altimetry or imagery. In essence, the HydroSat time series are complementary to the existing database. Existing databases like Hydroweb or DAHITI would provide time series for a limited set of lakes and rivers, which also holds for HydroSat. In fact, none of the available databases cover the entire water cycle. HydroSat complements the existing databases on many rivers and lakes, many of which are ungauged. We have made these points clear in the text.

Additionally, the literature is not up-to-late, which also somehow prevents the understanding of the novelty. See my specific comments below.

We addressed your comments and updated the literature.

Line 10: "..act as inputs to hydrological models". Hydrologic models generally use climate forcing data, terrain and land cover data as inputs. How this dataset can be used as the inputs is not clear to me.

Well, here we generally mean the fact that a hydrological model may itself be calibrated to or otherwise constrained by the provided quantities of HydroSat, or may incorporate them as input or via data assimilation. We rephrased the sentence to keep this generality:

*They can be incorporated for hydrological modelling..*

Line 25: not clear about what's known vs unknown. At which spatial and temporal scales?

It is clarified.

 Introduction section

global water cycle is a big topic. The current view is not extensive enough. I would recommend incorporate the insights of existing papers on this topic. One example is given below:

Rodell, M., Beaudoing, H. K., L'ecuyer, T. S., Olson, W. S., Famiglietti, J. S., Houser, P. R., ... & Wood, E. F. (2015). The observed state of the water cycle in the early twenty-first century. Journal of Climate, 28(21), 8289-8318.

We have included a paragraph in the introduction to give a better insight, but as you have rightly said, it is a big topic and cannot be discussed in detail here.

A review of existing products is missing. Without it, it would be difficult to understand the need of a new product.

At the beginning of each section you will find a detailed review of the existing products. We would avoid listing them again in the introduction, as the paper is already long enough. However, we made a passage in the introduction section mentioning a couple of products and characterised HydroSat quantities.

Line 105: the difference between SR and HR products is not clear. Please clarify

Line 110: improved temporal resolution compared to what? What's the exact temporal resolution? Does the dataset cover all lakes and rivers or a subset? If a subset, any filtering steps on lakes and rivers?

We clarified this in the text. An SR water level time series is the basic altimetry product of HydroSat with a temporal resolution given by the repeat period of the altimetry (e.g. 35 days for Envisat). It is the input to algorithms which provide HR water level time series over lakes, reservoirs, and rivers. The HR products come with an improved temporal resolution relying on multi-mission altimetry for both lakes and rivers.

Line 258: There are more studies on generating area time series from Landsat, such as

"Yang, K., Yao, F., Wang, J., Luo, J., Shen, Z., Wang, C., Song, C., 2017. Recent dynamics of alpine lakes on the endorheic Changtang Plateau from multi-mission satellite data. J. Hydrol. 552, 633–645.

Yao, F., Wang, J., Wang, C., Crétaux, J.-F., 2019. Constructing long-term high-frequency time series of global lake and reservoir areas using Landsat imagery. Remote Sens. Environ. 232, 111210."

Indeed. Thanks for introducing these relevant studies. We cited both in the text.

Line 115: I appreciate the reported validations for individual cases. But a global-scale evaluation makes more sense to me. Have you compared the coverage (spatially and temporally) and accuracy with existing products?

Line 125: same comment as above for the lake products

Yes. That would make more sense to us too. However, the lack of in situ data makes performing statistical analysis of all available data from all available databases/repositories impossible. Moreover, the geographic distribution of virtual stations offered by different data providers is also very different. Therefore, a global-scale evaluation is not possible at all.

Table 2: this list does not reflect the up-to-late status. A more comprehensive literature review is required. Just name a few excluded studies:

"Pickens, A. H., Hansen, M. C., Hancher, M., Stehman, S. V., Tyukavina, A., Potapov, P., ... & Sherani, Z. (2020). Mapping and sampling to characterize global inland water dynamics from 1999 to 2018 with full Landsat time-series. Remote Sensing of Environment, 243, 111792."

Yao, F., Wang, J., Wang, C., Crétaux, J.-F., 2019. Constructing long-term high-frequency time series of global lake and reservoir areas using Landsat imagery. Remote Sens. Environ. 232, 111210.

Zhao, G., Gao, H., 2018. Automatic correction of contaminated images for assessment of reservoir surface area dynamics. Geophys. Res. Lett.

Table 2 lists the data sources that provide time series of surface water extent and it is not meant to be a list of the algorithms or studies. In the revised manuscript, we have added the mentioned studies in the text.

Line 363: The original resolution of GRACE is 3 degree. How you downscaled the resolution should be introduced. Any cautions should be paid when using this downscaled product?

We did not downscale the results. We synthesise the spherical harmonics on a half by half degree grid cell. Unlike imagery, GRACE does not have a fix resolution. The resolution is rather signal strength dependent.

Table 4: same comment as on Table 2. For example,

"Yao, F., Wang, J., Yang, K., Wang, C., Walter, B.A., Crétaux, J.-F., 2018. Lake storage variation on the endorheic Tibetan Plateau and its attribution to climate change since the new millennium. Environ. Res. Lett."

"Wang, J., Song, C., Reager, J.T., Yao, F., Famiglietti, J.S., Sheng, Y., MacDonald, G.M., Brun, F., Schmied, H.M., Marston, R.A., Wada, Y., 2018. Recent global decline in endorheic basin water storages. Nat. Geosci. 11, 926–932."

Both studies have been added to Table 4.

Line 405: I would apologize if I missed anything. How you estimated the uncertainty of the storage anomaly?

In the case of the terrestrial water storage anomaly, the calibrated error of the GRACE data was used to obtain the uncertainty through error propagation. In the case of the lake water storage anomaly, our products do not include uncertainty. This is limited because estimating a proper uncertainty for the surface water extent needs a comprehensive understanding of various sources of the errors and uncertainties.

Conclusion

The conclusion seems to be abstract. For example, how many lakes and rivers have been covered in the dataset? I expect to see more quantitative summaries and highlights on the improvements upon existing products.

We provided this info in the "Data availability" section. A snapshot of all data (taken in April 2021) with a total of 10810 time series: 34 time series on surface water extent, 1323 time series on water level, 36 time series on river discharge, and 463 time series on water storage anomaly are available in GFZ data services. The highlights of the products are listed in the conclusion section.